# FIDELITY BREEDS COMPLEXITY: SIMULATING STOCK MARKETS WITH LARGE-SCALE GENERATIVE AGENTS

## ABSTRACT

Stock markets are one of the most complex systems in the modern world, where prices emerge from billions of decentralized interactions among heterogeneous participants in an ever-evolving information landscape. Building a high-fidelity stock market simulator is not only a cornerstone for understanding such complexity, but also offers a valuable testbed for anticipating and mitigating crises and disruptions. Despite decades of efforts, existing methods remain confined to an unresolved dilemma: structural fidelity often comes at the cost of non-intelligent agents, while large language model (LLM) agents can only participate in oversimplified market environments. To this end, we propose MarketSim, a large-scale stock market simulation framework with generative agents. Specifically, we first design a hierarchical multi-agent architecture. By decoupling agents' strategic reasoning from their high-frequency actions, this architecture enables LLM agents to participate in a nanosecond-resolution, NASDAQ-like continuous double auction market. Building on this, we simulate over 15k diverse market participant agents, whose billions of interactions collectively create an evolving market environment in which agents learn from feedback and adapt their strategies accordingly. Furthermore, we ground these agents in a rich informational landscape that covers over 12k real-world news articles, policy documents, and earnings reports. To evaluate our proposed MarketSim, we develop a comprehensive benchmark that includes stocks from 8 GICS sectors and 3 representative real-world scenarios, along with 5 stylized facts for market complexity and 5 price-related statistical metrics. Extensive experiments demonstrate that MarketSim not only captures the complexity characterizing real-world markets, but also accurately tracks real-world high-frequency price dynamics with an average MAPE of 3.48%. Overall, MarketSim not only offers direct applications in understanding and anticipating financial crises, but also provides evidence for a key tenet of complexity science: fidelity breeds complexity.

## 1 INTRODUCTION

Stock markets are the nerve center of the global economy. Despite being powerful engines of economic growth (Levine, 1997), they are often the source of abrupt and widespread systemic collapses (Farmer & Foley, 2009; McKee & Stuckler, 2020). Throughout history, stock markets have experienced countless instabilities such as black swan events, herd behavior, and volatility clustering, all of which reflect the complexity that governs market behavior. In recent years, market–triggered crises have become more frequent, ranging from global disruptions such as the 2008 financial crisis (Farmer & Foley, 2009), to liquidity breakdowns driven by emergencies (McKee & Stuckler, 2020), to flash crashes triggered by policies like the Liberty Day tariff (Wikipedia, 2025). On the other hand, facing such crises, we are at a loss due to the lack of a high-fidelity stock market simulator that can help us understand and anticipate their dynamics.

However, building such a simulator is a non-trivial task. This is primarily because stock markets are complex adaptive systems, where collective outcomes, *i.e., prices*, emerge from decentralized, nonlinear interactions among large numbers of heterogeneous agents. To decode this complexity, researchers have made efforts to develop agent-based models (ABMs) that aim to replicate stock markets (Arthur et al., 2018; Byrd et al., 2020; Belcak et al.; Axtell & Farmer, 2025). Among them, the ABIDES platform (Byrd et al., 2020) stands out for enabling thousands of simple agents to

trade under the continuous double auction (CDA) mechanism, capturing the high-frequency price dynamics in stock markets. While these models largely preserve structures of stock markets, their agents lack ***behavioral fidelity***: driven by pre-defined heuristics or rule-based strategies, such agents fail to capture how real-world market participants perceive, interpret, and respond to information. More importantly, this low fidelity in agent behavior further prevents them from grasping one of the stock market's core mechanisms, *i.e.,* that price changes arise from the collective responses of participants to new information (Fama, 1970; Axtell & Farmer, 2025).

Recent advances in large language models (LLMs) have shown the potential to improve behavioral fidelity (Park et al., 2023; Gao et al., 2024a; Li et al., 2024b). Some researchers have begun to explore replacing traditional agents with LLM-driven ones in stock market simulations (Yang et al., 2025; Gao et al., 2024b; Zhang et al., 2024). However, as a trade-off for current limited agent designs, they typically oversimplify key structures of stock markets. For example, they adopt turn-based trading schemes that fundamentally violate CDA, thereby distorting the essential price discovery process that defines market behavior. Furthermore, they often simulate a small number of agents with oversimplified labels, such as "aggressive" or "conservative", misrepresenting the scale of real-world participants and their decision-making processes (Brav et al., 2024; Blume et al., 2017). To sum up, their low ***structural fidelity*** hinders them from reproducing the complex emergent dynamics of real-world markets.

These two lines of studies lead to a central question: what makes up a high-fidelity simulator of stock markets? Specifically, how can we design a simulator that captures both behavioral and structural fidelity? To this end, we propose MarketSim, a high-fidelity LLM-empowered nano-scale stock Market Simulation framework. Specifically, we begin by modeling institutional investors, who account for the majority of trading volume in real-world markets and have highly complex decision-making processes (Brav et al., 2024; Blume et al., 2017). We propose a hierarchical, LLM-empowered multi-agent architecture inspired by organizational logic behind real-world institutions. In each simulated institution, two distinct types of agents cooperate: reasoning on the current information landscape, fund manager agents formulate instructions on investment strategies and indicative prices; trader agents are dynamically configured to execute high-frequency trades in managers' instructions. In this way, we enable agents with institutional-level intelligence to trade in NASDAQ-like stock markets in nanosecond resolution. After constructing the internal world of agents, we focus on situating them in a rich environment where they can autonomously evolve. The environment consists of two facets: One is collectively built by over 15k institutional and background agents (e.g., retail investors and market makers), emerging from their billions of trading interactions. These interactions generate prices, returns, and losses, which provide agents with meaningful feedback, in turn shaping their future strategies. The other facet is the real-world informational landscape, where we incorporate a massive corpus of over 12k news articles, policy documents, and corporate financial reports.

To evaluate MarketSim, we design a comprehensive benchmark that covers stocks from 8 GICS Level-1 sectors (e.g., Energy, Information Technology), across three representative real-world scenarios: the Liberal Day tariff shock, DeepSeek's market debut, and corporate earnings announcements. Moreover, we incorporate 5 stylized facts that qualitatively characterize well-known market complexity, along with 5 price-related statistical metrics that quantitatively measure the alignment between real-world and simulated stocks. Extensive experiments demonstrate that MarketSim faithfully reproduces all five key facts, suggesting that the simulated market exhibits realistic complexity, ranging from black swan events and herding behavior to short-term uncertainties and long-term regularities. Moreover, MarketSim accurately tracks high-frequency price dynamics observed in real-world markets, as validated by five well-established quantitative metrics with an average MAPE of 3.48%. Ablation studies confirm that removing any designs for behavioral or structural fidelity substantially degrades the system. Overall, our work paves the way for a new generation of high-fidelity stock simulators, offering a powerful computational testbed for understanding, anticipating, and ultimately curbing financial crises. Our contributions can be summarized into three folds:

- We propose the first high-fidelity stock market simulation framework with generative agents.
- We design a hierarchical multi-agent architecture, which enables agents to participate in NASDAQ-like high-frequency trading.
- We introduce a comprehensive benchmark for stock market simulations, covering stocks from 8 diverse sectors, 3 representative scenarios, 5 stylized facts for market complexity, and 5 price-related statistical metrics.

## 2 RELATED WORKS

We review three lines of related work: (i) stock market and its complexity, which characterizes the object of our modeling; (ii) traditional agent-based modeling, which outlines established modeling approaches; and (iii) large model-based simulations, which reflect recent progress in LLM-driven market modeling.

**Stock Market and its Complexity.** While classical financial theories, such as the Efficient Market Hypothesis, posit that prices should converge to a stable equilibrium (Fama, 1970), extensive empirical evidence reveals a different reality. Real-world markets consistently exhibit stylized facts, including fat-tailed returns and volatility clustering (Cont, 2001), as well as non-equilibrium phenomena such as price bubbles and crashes. These persistent deviations suggest that markets are not simple equilibrium-seeking systems, but rather complex adaptive systems, where macro-level patterns emerge from decentralized micro-level interactions (Arthur, 1995). At the heart of this complexity lies a core micro-level mechanism: the CDA, populated by large numbers of heterogeneous, boundedly rational agents. The collective, adaptive expectations of these agents, formed in response to an ever-evolving stream of endogenous and exogenous information, lead to persistent changes in price. Overall, modeling stock markets requires adopting a complexity perspective and faithfully replicating the structural and behavioral dynamics of real-world markets.

**Traditional Agent-Based Modeling.** Given the market's nature as a complex adaptive system, the ABM paradigm emerged as a natural bottom-up approach to study it (Farmer & Foley, 2009). Pioneering works like the Santa Fe Artificial Stock Market demonstrate the promise of this approach. In this model, agents using simple rules and genetic algorithms to adapt their trading decisions successfully replicated several stylized facts observed in real markets (Palmer et al., 1999; Arthur et al., 2018). Subsequent research further explores the importance of agent intelligence (Capterra, 2019; Manahov et al., 2014); for example, Manahov et al. (2014) show that agent cognitive ability significantly impacts market characteristics. More recent studies like ABIDES represent a significant leap in structural fidelity, offering an open-source simulation of Nasdaq-like markets (Byrd et al., 2020). Despite these advances, a critical gap remains. Traditional agents lack behavioral fidelity: they rely solely on structured order-flow data while ignoring crucial unstructured signals such as news, policy changes, or market sentiment. Moreover, their decision-making processes are exogenously specified and overly simplistic (Friedman, 2018), failing to capture the nuanced reasoning and strategic adaptability of human traders.

**Large Model-Based Simulation.** The advent of LLMs offers a promising solution to the behavioral fidelity gap in traditional ABMs, enabling agents to perceive, interpret, and respond to complex information (Yu et al., 2024; Xiao et al., 2024). Several studies have integrated LLM-driven agents into simulations to generate more human-like behaviors (Yang et al., 2025; Gao et al., 2024b; Zhang et al., 2024). However, the emphasis on human-likeness often comes at the cost of structural fidelity. Key market mechanisms are frequently oversimplified, for instance, by degrading the CDA to turn-based interactions and reducing the market's scale and heterogeneity to a few agents with simplistic archetypes. More recently, a data-driven large market model, MarS, has been proposed, which "flattens" diverse market participants into a single generative model to simulate order books at scale (Li et al., 2024a). Although quantitatively accurate, this black-box model lacks a mechanistic foundation, which makes it difficult to capture and interpret emergent phenomena, particularly in unprecedented scenarios like the 2008 financial crisis, where simulation becomes most valuable (Farmer & Foley, 2009).

Overall, the above three lines of work underscore a key yet unresolved challenge: Modeling complex stock markets requires a unified framework that captures both **structural and behavioral fidelity**, which is the central contribution of our work.

## 3 FRAMEWORK

To address the challenge of achieving both behavioral and structural fidelity, we introduce MarketSim, a LLM-empowered stock market simulation framework. As illustrated in Figure 1, the framework is organized into three hierarchical scales: the micro level, which defines the participant agents; the meso level, which delineates the market structures; and the macro level, which constitutes the information landscape. We will elaborate on each of them in the following sections.

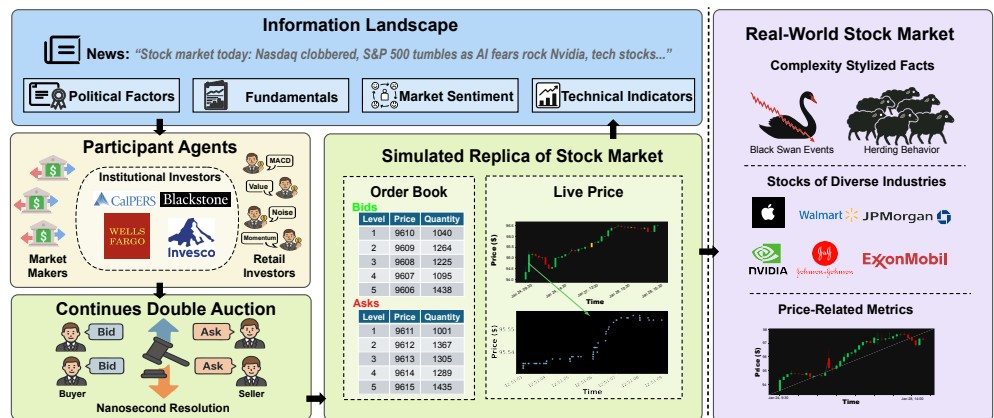

Figure 1: Overview of the proposed framework MarketSim, which captures the complexity of stock markets through three scales.

## 3.1 MICRO: PARTICIPANT AGENTS

The micro level of MarketSim populates the simulation with a heterogeneous agent population with high behavioral fidelity. First, we focus on institutional investors, who not only account for the majority of trading volume but also display complex, information-based decision-making processes in real-world markets (Brav et al., 2024; Blume et al., 2017). To capture their complex reasoning while preserving high-frequency trading capabilities, we design a hierarchical agent architecture empowered by LLMs. Second, we incorporate a rich ecosystem of background agents, such as heuristic-driven retail investors and liquidity-providing market makers.

### 3.1.1 INSTITUTIONAL AGENTS

Our institutional agent adopts a hierarchical two-level architecture to address a fundamental trade-off. While LLMs offer the deep reasoning capabilities necessary for high behavioral fidelity, they are too slow and computationally intensive for high-frequency market interactions. To resolve this, we draw inspiration from real-world investment institutions, where fund managers make low-frequency, information-rich strategic decisions, and traders focus on executing those strategies at high speed and optimal prices (Golec, 1996; Cohen et al., 2005). This division of labor enables institutions to operate effectively in dynamic markets. Therefore, following this real-world division, we separate the strategic "brain" from the tactical "hands": a high-level manager agent, empowered by an LLM, handles complex low-frequency decisions, while a team of low-level trader agents rapidly executes the resulting decisions at nanosecond speed. Formally, the overall of an institution $i$, denoted as $\pi_{\text{inst}}^{(i)}$, is a composition of its manager's policy $\pi_{\mathcal{H}}^{(i)}$ and the policies of its $K$ individual traders $\{\pi_{\mathcal{L}}^{(j)}\}_{j=1}^{K}$.

**The Manager Agent $\mathcal{H}$.** The manager agent acts as the strategic core of the institution, designed to simulate the cognitive process of a real-world fund manager. As shown in Figure 2, its cognitive architecture comprises four key components: an empirically-grounded profile, dynamic memories encompassing both short- and long-term storage, and accumulated experience.

First, the agent's cognitive process begins by perceiving the information landscape, drawing on a rich information set $I_t$ from both exogenous sources, such as news and policy signals, and endogenous signals, such as the live order book. This raw data is distilled into short-term memories $\mathcal{M}_{\text{ST},t}$, which capture the agent's real-time awareness of the present market state. These memories include sentiments derived from external news and policies, as well as technical indicators from market data.

This real-time perception is subsequently consolidated into the agent's knowledge base, stored as long-term memory $\mathcal{M}_{\text{LT},t}$. This knowledge accumulates through two pathways. Salient short-term memories, e.g., the enduring economic impacts of an initial news shock, are periodically summarized and transferred into long-term memory. In parallel, fundamental information like corporate earnings reports, which indicate a firm's underlying financial health, is directly encoded into this long-term knowledge base.

Beyond interpreting external data, the agent also learns from its own actions through self-reflection. By evaluating market feedback, e.g., the profitability of past strategies, it updates its experience

Figure 2: A hierarchical multi-agent architecture for simulating institutional investors.

$\mathcal{E}_t$, enabling continual adaptation of its decision-making policy. Finally, these dynamic cognitive processes are all moderated by the agent's intrinsic profile. Each manager agent is initialized with a unique, empirically-grounded profile $\mathcal{P}^{(i)}$ that delineates its investment style. This process is formally represented by defining the agent's comprehensive internal state:

$$A_{H,t}^{(i)} = (\mathcal{P}^{(i)}, \mathcal{M}_{ST,t}^{(i)}, \mathcal{M}_{LT,t}^{(i)}, \mathcal{E}_t^{(i)}). \tag{1}$$

The agent's policy $\pi_H$ transforms $A_{H,t}^{(i)}$ into a trading guidance signal $G_t$:

$$G_t = \pi_H(A_{H,t}^{(i)}) = (\mu_t, \mathbf{w}_t), \tag{2}$$

where $\mu_t$ is an indicative price, and $\mathbf{w}_t$ is a vector of weights determining the allocation among different trading strategies for its subordinate trader agents.

**The Trader Agents $\mathcal{L}$.** The trader agents are the high-frequency execution arm of the institution, representing individual traders who act on the manager's guidance. They are lightweight, computationally efficient agents that receive the strategic guidance tuple $G_t = (\mu_t, \mathbf{w}_t)$ from their high-level manager agent. Upon receiving this guidance, each trader agent $j$ executes a two-step process: policy selection and parameterized execution.

First, the agent selects its trading policy for the current period. The manager's weight vector $\mathbf{w}_t$ acts as a probability distribution over a predefined set of available strategies $\Pi_L$ (e.g., value-based, momentum-based). The trader agent samples its current strategy, $\pi_{\mathcal{L},t}^{(j)}$, from this categorical distribution:

$$\pi_{\mathcal{L},t}^{(j)} \sim \text{Categorical}(\Pi_L, \mathbf{w}_t). \tag{3}$$

Second, the agent executes its chosen policy in the current period. The indicative price $\mu_t$ serves as a key parameter for policies that require a reference price, such as value-based trading. The final action $a_{j,t}$ (e.g., submitting an order) is thus a function of the current market state $X_t$, conditioned on the parameters derived from the manager's guidance:

$$a_{j,t} = \pi_{\mathcal{L},t}^{(j)}(X_t; \mu_t). \tag{4}$$

Overall, this practice-inspired hierarchical design allows MarketSim to model both the deep, information-driven reasoning of institutions and their rapid, real-time impact on the market.

### 3.1.2 BACKGROUND AGENTS

To create a realistic market ecosystem for our institutional agents to interact with, we populate the simulation with a diverse population of background agents. While these agents represent a minority of the trading volume, they are crucial for a complete market environment. Here, we focus on two primary categories: retail investors and market makers (Easley & O'Hara, 1995).

**Retail Agents.** We model retail investors along a spectrum of intelligence. At the simplest level, noise agents emulate the random behavior of uninformed traders; they are activated once per day following a U-quadratic distribution and submit random market orders (Graczyk & Duarte Queiros, 2016). At an intermediate level, momentum agents operate as heuristic-driven trend followers, making decisions based on moving-average indicators derived from high-frequency price data. At the highest level, value agents represent investors conducting pseudo-fundamental analysis, such as inferring value from institutional research reports. Accordingly, we assume they trade based on an estimated fundamental value, computed as the average of indicative prices proposed by all institutional agents, with a variance term added to capture heterogeneity and idiosyncratic noise.

**Market Maker Agents.** To ensure market liquidity and realistic price dynamics, we include agents that emulate the role of market makers (Easley & O'Hara, 1995). These agents provide liquidity by maintaining both bid and ask orders. They employ an adaptive strategy, dynamically adjusting their quote prices, depths, and spreads in response to real-time market trading volume and volatility, thereby approximating the behavior of liquidity providers in real financial markets.

## 3.2 MESO: MARKET STRUCTURES

After establishing the micro-level agent populations, we now define the meso-level market structure that governs their interactions. To achieve high structural fidelity, we design the market environment as an asynchronous, event-driven system operating under CDA mechanisms. This design follows established practices in the ABIDES simulator and aligns with real-world NASDAQ protocols (Byrd et al., 2020). To ensure the temporal integrity required by the CDA, the simulation is built on an event-driven architecture modeled after NASDAQ protocols. All market interactions are encapsulated as discrete, time-stamped messages that are processed in strict chronological order, with nanosecond-level resolution. This design guarantees causal consistency, ensuring that events are handled precisely as they occur in simulated time.

The order matching process from an agent's decision to a potential trade follows a precise lifecycle. It begins when an agent generates an order, defined as a tuple specifying its action, price, quantity, and timestamp. Once this timestamp is reached, the order is processed by a central matching engine that maintains the Limit Order Book (LOB). The engine attempts to match the incoming order against resting orders based on strict price-time priority. A trade is executed once the matching condition is satisfied, and the execution price is set to the prevailing market price at that moment under the CDA mechanism (please check formal formulation in Appendix N). Any unfilled portion of a new order is added to the LOB, and a confirmation message detailing the outcome is subsequently sent to the originating agent.

## 3.3 MACRO: INFORMATION LANDSCAPE

The macro-level foundation of MarketSim is the information landscape, which underpins all agent decision-making. This landscape comprises two distinct types of information: endogenous information, generated within the simulated market, and exogenous information, sourced from real-world data and events. These two components play complementary roles: endogenous information maintains internal coherence and dynamic feedback within the simulation, while exogenous information anchors agent behavior to external realities, ensuring relevance to actual market narratives and shocks.

**Endogenous Information.** Endogenous information reflects the real-time internal state of the market, derived primarily from the order book. All agents can query the market structure to access a stream of structured data points, including current bid-ask spreads, market depth, and midpoint prices. This information allows agents, particularly those driven by technical rules, to form perceptions of the market's immediate liquidity, volatility, and short-term trends.

**Exogenous Information.** Exogenous information grounds the simulation in real-world scenarios. To this end, we collect and inject a corpus of real-world data aligned with the simulation period, including news articles, major policy announcements, and corporate earnings reports. This rich and often unstructured information is crucial for the LLM-driven manager agents, enabling them to develop nuanced, human-like perceptions of firms' fundamental values, relevant political dynamics,

and overall market sentiment. As a result, the simulation can respond to the same external events that shape real-world market behavior.

# 4 EXPERIMENTS

To systematically evaluate the proposed MarketSim framework, we design and conduct a series of experiments centered on the following three research questions (RQs), each probing a critical aspect of the system: realism, accuracy, and generalization. To further validate our design, we conduct an ablation study that examines the contribution of each key component in the proposed MarketSim.

- **RQ1. Qualitative Realism:** Can MarketSim reproduce the well-established stylized facts that capture the complexity of real-world stock markets?
- **RQ2. Quantitative Accuracy:** How closely do the price dynamics generated by MarketSim align with real-world data, as measured by a suite of quantitative metrics?
- **RQ3. Generalization:** Can MarketSim generalize across varying market conditions, such as different industrial sectors and diverse types of real-world news events?

**Experimental Benchmark**

*Stocks and Scenarios.* We design a comprehensive evaluation benchmark covering a diverse set of stocks and shock scenarios to rigorously test the capabilities of MarketSim. Our stock selection spans eight distinct GICS Level-1 sectors (i.e., Information Technology, Communication Services, Consumer Staples, Healthcare, Financials, Industrials, Energy, and Utilities), ensuring that evaluations are not limited to a single industry. To mitigate the risk of LLM data leakage, we choose three real-world shock events from late 2024 to early 2025: (i) the "Liberal Day" tariff, representing a global policy shock; (ii) DeepSeek's market debut, reflecting a sentiment-driven shock; and (iii) corporate earnings announcements, capturing fundamental information disclosures. Each scenario is grounded in a rich corpus of real-world data, including over 12k news articles, financial reports, and policy releases, sourced from Finnhub, Bloomberg, Newsdata.io, Wind, and FactSet. A key feature of our experimental design is its emphasis on heterogeneity. For each scenario, we deliberately include stocks with varied real-world responses. For example, during the tariff shock, globally exposed firms like Apple are heavily affected, while less globally exposed firms like Johnson & Johnson remain relatively insulated. This setup allows us to assess not only whether MarketSim can reproduce general market trends, but also whether it can capture nuanced, firm-specific dynamics. Our primary large language model is DeepSeek R1, with Qwen3-8b and Llama-3.1-8b used for generalizability experiments. Detailed configurations, including selected stocks and data sources, are provided in Appendix X.

*Baselines.* To rigorously evaluate the effectiveness of MarketSim, we introduce two classes of baselines: (i) predictive models, including autoregressive methods (Moving Average, ARIMA), traditional machine learning models (Linear Regression, LightGBM), and deep learning models (LSTM, Transformer). These models are trained either on the preceding week of price history (for autoregressive models) or on the preceding week of prices combined with news embeddings (for the higher-capacity models); and (ii) ABM baselines constructed by integrating the same predictive models into the ABIDES framework Byrd et al. (2020), replacing its exogenous indicative price, which in ABIDES is normally derived from the real-world price series, with model-generated values. This replacement avoids the unfairness of comparing MarketSim's fully endogenous reasoning with a simulator that relies on real-world price trajectories as external guidance. It is worth noting that this comparison is inherently conservative for MarketSim, as prediction and simulation are fundamentally different tasks. Predictive baselines function as curve-fitting models that treat the market as a black box and optimize numerical forecasts, whereas MarketSim performs mechanism generation by reconstructing the underlying agent interactions that produce realistic price dynamics.

*Qualitative Realism via Stylized Facts.* To assess the qualitative realism of our simulation (RQ1), we evaluate its ability to reproduce five core stylized facts that characterize the emergent complexity of real financial markets (Cont, 2001). These facts capture the market's dual nature of short-term unpredictability and long-term structure. Non-stationarity, where price series exhibit unit root characteristics reflecting their random walk nature, and the absence of linear autocorrelation in returns together imply that future prices cannot be predicted from historical price information alone. Yet

Table 1: Performances of MarketSim and all other baselines on JNJ in Liberal Day tariff shock.

| | Category | Autoregressive | | Traditional ML | | Deep Learning | | ABM | | | | | | Ours |
|---|---|---|---|---|---|---|---|---|---|---|---|---|---|---|
| | Model | MA | ARIMA | Linear | LightGBM | LSTM | Trans. | Linear | MA | ARIMA | LightGBM | LSTM | Trans. | MarketSim |
| **Qual.** | Abs. of L.A. | × | × | × | ✓ | × | × | ✓ | ✓ | ✓ | ✓ | ✓ | × | ✓ |
| | Fat Tails | ✓ | ✓ | ✓ | ✓ | ✓ | ✓ | ✓ | ✓ | ✓ | ✓ | × | ✓ | ✓ |
| | Agg. Gauss. | ✓ | ✓ | ✓ | ✓ | ✓ | ✓ | × | ✓ | × | ✓ | ✓ | ✓ | ✓ |
| | Volat. Clust. | ✓ | ✓ | ✓ | × | × | ✓ | × | × | × | ✓ | × | ✓ | ✓ |
| | Non-Stat. | × | × | ✓ | × | ✓ | ✓ | × | × | × | × | × | ✓ | ✓ |
| **Quan.** | RMSE | 3.833 | 4.293 | 1.809 | 3.808 | 3.314 | 6.459 | 1.816 | 3.830 | 4.275 | 3.792 | 3.327 | 6.497 | **1.614** |
| | MAPE (%) | 1.907 | 2.263 | 1.067 | 1.895 | 1.815 | 3.246 | 1.071 | 1.904 | 2.250 | 1.882 | 1.820 | 3.264 | **0.816** |
| | DTW Distance | 0.490 | 0.525 | 0.291 | 0.507 | 0.609 | 0.153 | 0.014 | 0.030 | 0.030 | 0.027 | 0.026 | **0.005** | 0.011 |
| | Q-Q Corr. | 0.482 | 0.446 | 0.892 | 0.432 | 0.659 | 0.817 | 0.988 | 0.988 | 0.989 | 0.986 | 0.981 | **0.995** | 0.993 |
| | Volatility Sim. | 0.012 | 0.035 | 0.025 | 0.247 | 0.022 | 0.156 | 0.433 | 0.404 | 0.379 | 0.464 | 0.412 | 0.556 | **0.796** |

markets deviate from pure randomness: fat-tailed return distributions indicate that extreme price movements occur more frequently than predicted by normal distributions, reveal higher probabilities of extreme "black swan" events than normal distributions suggest. Volatility clustering shows periods of high and low volatility tend to persist, attributed to information clustering,investor sentiment, and collective behavioral patterns such as herding. Over longer horizons, aggregated Gaussianity emerges as return distributions converge toward normality with increasing time scales, suggesting that fundamental drivers and arbitrage mechanisms gradually dominate market dynamics. We verify these properties through unit root tests for non-stationarity, ACF analysis for autocorrelation, kurtosis evolution across time scales for aggregated Gaussianity, GARCH models for volatility clustering, and Q-Q plots combined with kurtosis tests for fat tails.

***Quantitative Accuracy via Statistical Metrics.*** To quantitatively assess the alignment between the simulated price series (RQ2), we employ a collection of five statistical metrics. We begin by evaluating the direct time-series similarity of prices. We use (i) Root Mean Squared Error (RMSE) and (ii) Mean Absolute Percentage Error (MAPE) to measure point-wise accuracy. To capture morphological similarity, we adopt (iii) Dynamic Time Warping (DTW) Distance, which is robust to temporal shifts and distortions between the two series. Moving beyond the price series itself, we assess the distributional similarity of returns using (iv) Q-Q Correlation, which measures the linear correlation of the series' quantiles. Finally, to evaluate the alignment of volatility characteristics, we use **(v) the volatility similarity score**, a composite metric that measures similarity across three dimensions: the magnitude of daily price movements, the frequency of significant price changes, and the rate of trend reversals. Details on the evaluation procedures for these stylized facts and quantitative metrics are provided in the Appendix C.

### 4.1 EXPERIMENTAL RESULTS

**RQ1: Qualitative Realism.** Our analysis reveals that MarketSim successfully reproduces all five stylized facts across the full range of experiments, covering 12 stocks from 8 GICS sectors and all 3 shock scenarios (summarized in Table S6). This consistent result has two key implications. First, it validates our selected benchmarks, confirming that these stylized facts are indeed universal properties of the empirical data. Second, and more importantly, it demonstrates that by ensuring both structural and behavioral fidelity, MarketSim can capture the emergent complexity of real-world markets—from "black swan" events (Figs. S6-S9, S12-S15, S18-S21b&c) to herding behaviors (Fig. 3), and from short-term uncertainties (Figs. S6-S9, S12-S15, S18-S21a) to long-term structure. As shown in Tabs. 1 and S9, we find that across all baselines, including predictive models and ABMs, none are able to reproduce the full set of five stylized facts. Each model captures only partial statistical properties and fails to fully capture the complexity in real markets.

**RQ2: Quantitative Accuracy.** To address RQ2, we quantitatively assess the alignment between our simulated price series and the real-world data across all scenarios. The results, presented in Tabs. 2, S7 & S8, demonstrate that MarketSim achieves a high degree of quantitative accuracy. In terms of direct time-series similarity, the model shows strong performance with an average MAPE of 3.48% and a consistently low DTW distance, indicating high morphological similarity. Furthermore, the model captures deeper statistical properties with high fi-

Table 2: Results of Liberal Day tariff Shock.

| Metrics | AAPL | JNJ | JPM | XOM |
|---|---|---|---|---|
| RMSE | 16.485 | 1.614 | 6.655 | 0.992 |
| MAPE (%) | 5.856 | 0.816 | 2.705 | 0.720 |
| DTW Distance | 0.009 | 0.011 | 0.005 | 0.003 |
| Q-Q Corr. | 0.988 | 0.993 | 0.994 | 0.993 |
| Volatility Sim. | 0.618 | 0.796 | 0.446 | 0.787 |

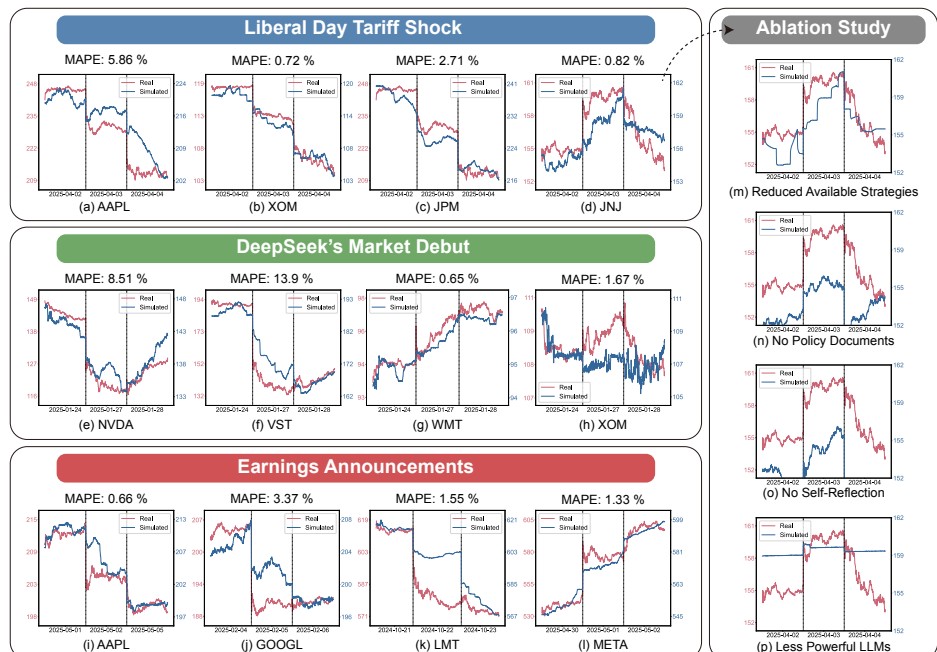

Figure 3: Comparison between real-world and simulated stock data across 12 stocks from 8 GICS sectors and all 3 shock scenarios.

delity. The Q-Q Correlation remains exceptionally high across all twelve experiments (all values > 0.97), signifying a near-perfect alignment of the simulated and real return distributions. The volatility similarity score also shows strong results, confirming that the model effectively reproduces the complex volatility of the real market.

A deeper analysis reveals that the model's accuracy is correlated with the magnitude of the price shock, a valuable insight into its current capabilities. In scenarios with moderate volatility, such as for Johnson & Johnson (JNJ) during the tariff event or Apple (AAPL) during its earnings release, the model's performance is exceptionally strong, with MAPE values as low as 0.82% and 0.66%, respectively. However, for stocks experiencing extreme, outsized shocks, such as Vistra Corp. (VST) during the DeepSeek debut, which saw a real-world drop of nearly $40, the model captures the correct downward trend but underestimates the full magnitude of the collapse, resulting in a relatively higher RMSE of 23.85 and MAPE of 13.95%.

By comparing MarketSim with all baselines (Tab. 1 and S9), we find that MarketSim achieves the most consistent and comprehensive quantitative performance across stocks and scenarios. While some baselines perform well on individual metrics, none maintain strong accuracy across the full evaluation suite. In contrast, MarketSim reduces RMSE by 11% to 41% and MAPE by 24% to 39% relative to the strongest alternatives, and improves volatility similarity by 15% to 43%. The only metric where a baseline occasionally attains a better value is DTW, shown by the transformer-based ABM variant. However, this model performs substantially worse on RMSE, MAPE, and volatility similarity. This is because DTW measures shape similarity while allowing for temporal misalignment. The transformer-based ABM fails to capture the natural reaction lag to news events. MarketSim, by simulating the cognitive process of information assimilation, generates more realistic timing in price responses.

**RQ3: Generalizablity.** To answer RQ3, we test the framework's ability to generalize across diverse stocks, sectors, and event types, with results visualized in Fig. 3. The findings confirm that MarketSim successfully captures a wide spectrum of nuanced, firm-specific market reactions. For instance, in response to the single Liberal Day tariff shock, the model captures both the sharp price decline in a trade-exposed firm like Apple (AAPL, Fig. 3a) and the distinct, inverted U-shaped trend of a domestically-focused firm like Johnson & Johnson (JNJ, Fig. 3d). The framework also reproduces other complex, non-linear patterns, such as the U-shaped drop-and-reversal of Nvidia (NVDA, Fig. 3e) during DeepSeek's market debut. Moreover, it correctly models the behavior of relatively unaffected stocks during the same shock, capturing the steady upward trend of Walmart

(WMT, Fig. 3g) and the volatile, sideways consolidation of ExxonMobil (XOM, Fig. 3h). By successfully modeling these varied dynamics—from sharp declines to complex reversals and sideways movements—across different industries and under diverse shocks, MarketSim demonstrates robust generalization and the ability to capture the heterogeneous responses that characterize real-world markets.

**Ablation Study.** To validate our design choices, we conducted a comprehensive suite of 17 ablation experiments. The findings consistently show that reducing the model's fidelity at either the agent or market level significantly degrades its ability to reproduce realistic market dynamics. First, we confirm the importance of behavioral fidelity. Replacing our empirically-grounded agent profiles with simplistic archetypes like "conservative" or "aggressive" increases MAPE from 0.82% to over 2.8% (Tab. S10). Similarly, degrading the manager's reasoning ability by using weaker LLMs substantially lowers its performance (Tab. S10), underscoring that sophisticated agent intelligence is crucial. This lack of behavioral fidelity is further highlighted when we remove key cognitive modules; for instance, ablating the self-reflection causes the simulation to fail in reproducing a key stylized fact and increases RMSE by over 10x (Tab. S12). Second, we validate the need for structural fidelity. Restricting the dynamic strategy allocation from the manager (Tab. S11) or adding disruptive market conditions, such as a surge of herd-like individuals or liquidity shocks (Tab.S13), shows that agent behavior is deeply shaped by the surrounding market structure. Overall, our ablation study confirms a central thesis: the emergent complexity of financial markets, from stylized facts to nuanced price movements, can only be captured when high behavioral and structural fidelity are jointly achieved. In short, fidelity breeds complexity.

**Realism of Agent Decisions.** To assess the realism of agent decisions, we have conducted a human evaluation study with 18 professional financial practitioners. Each expert evaluates a set of agent-generated decisions along three dimensions: market consistency, internal coherence, and decision soundness, using a 0–10 rating scale. Results show that the proposed agents receive high average scores across all three dimensions (7.32, 7.44, and 7.15), indicating that practitioners consider the generated decisions realistic and well-reasoned. We further validate this finding by constructing a matched control baseline, where each agent's decision is paired with a comparable decision from a similar price but unaffected by the shock event. Experts are asked to choose which of the two appears more realistic. Agent decisions are selected significantly more often (a binomial test, $p_0 = 0.745$, $p < 0.001$). Experts then rate both decisions independently along the three dimensions, and agent decisions obtain significantly higher scores in all cases (two-sided Student's $t$ tests, market consistency: $t = 7.09$, $p < 0.001$; internal coherence: $t = 5.72$, $p < 0.001$; decision soundness: $t = 6.91$, $p < 0.001$). These results show that MarketSim produces agent decisions that are consistently judged by domain specialists as realistic, coherent, and well-grounded.

**Applications.** We perform two experiments to show the potential of MarketSim as practical testbeds for understanding and anticipating shocks by additionally incorporating (i) 200 momentum-based agents who exhibit trend-chasing behavior, and (ii) agents that submit large-volume orders into the market. We observe that (i) market volatility increases (Fig. S29e), and (ii) liquidity depletion (Fig. S42), both patterns aligning with empirical observations in real-world markets.

## 5 CONCLUSION

In this paper, we introduce MarketSim, a simulation framework designed to resolve the critical trade-off between behavioral and structural fidelity in stock market modeling. Based on our proposed hierarchical multi-agent architecture, we demonstrate that MarketSim successfully reproduces a wide array of complex market dynamics, from emergent stylized facts to nuanced, firm-specific responses to real-world shocks. Furthermore, by simulating the market's response to disruptive conditions, such as sudden liquidity shocks and surges of herd-like trading, our ablation studies highlight the framework's potential as a powerful testbed for assessing financial risk. Our findings provide strong evidence that the emergent properties of stock markets are a product of this dual fidelity, underscoring a foundational principle for future research: ***fidelity breeds complexity***.

MarketSim provides a controlled environment for analyzing how market manipulation strategies may propagate and for evaluating the effectiveness of potential countermeasures. By enabling regulators and policymakers to stress-test market dynamics and detection algorithms, the simulator supports proactive identification of vulnerabilities before they appear in real markets.

## 6 REPRODUCIBILITY STATEMENT

To ensure the reproducibility of our results, all codes for MarketSim framework are available at `https://anonymous.4open.science/r/MarketSim-E854/`. Our primary large language model is DeepSeek R1, with Qwen3-8b and Llama-3.1-8b used for generalizability experiments. The composition of the agent population in our simulation is designed to mirror the participant structure of the real-world NASDAQ market (Brav et al., 2024; Blume et al., 2017) and prior practices (Byrd et al., 2020). Please check more details (e.g., specific prompts and computational resource) in Appendix.

## 7 ETHICS STATEMENT

No human participants are involved in this study, and no ethical issues are applicable.

## 8 USE OF LARGE LANGUAGE MODELS

After completing the initial draft, we use LLMs to polish the text and consult them on specific word choices.

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

## A DATASET

Table S3: Data Usage Summary

| Usage | Source | Data Type | Data Volume |
|---|---|---|---|
| | | Stock Price | 1.1M data points |
| | | Institutional Ownership | 960 holdings |
| Initialize Agent | Finnhub API, Wind, | Financials As Reported | 64 reports |
| Profiles | Bloomberg | SEC Filings | 64 filings |
| | | Social Sentiment | 3.2K scores |
| | | Technical Indicators | 39K indicators |
| Provide External | Finnhub, Newsdata.io, | News articles | |
| Information | Dow Jones Factiva | Policy announcements | 120K articles |

## B DETAILS ABOUT EXPERIMENTS

Table S4: Experimental Scenario Configurations

| Scenario | Simulation Period | Selected Stocks |
|---|---|---|
| Reciprocal Tariffs | Apr 2-4, 2025 | AAPL, JNJ, JPM, XOM |
| DeepSeek Shock | Jan 24-28, 2025 | NVDA, VST, WMT, XOM |
| Earnings Releases - AAPL 2025Q2 | May 1-5, 2025 | AAPL |
| Earnings Releases - META 2025Q1 | Apr 30 - May 2, 2025 | META |
| Earnings Releases - GOOGL 2024 | Feb 4-6, 2025 | GOOGL |
| Earnings Releases - LMT 2024Q3 | Oct 21-23, 2024 | LMT |

Table S5: Agent Configuration in Simulation Environment

| Agent Type | Quantity | Role/Description |
|---|---|---|
| Exchange Agent | 1 | Central market clearing and order matching |
| Noise Agent | 12,000 | Random traders simulating market noise |
| Value Agent | 50–100 | Fundamental value-based investors |
| Market Maker Agent | 4 | Provide liquidity and bid-ask spreads |
| Momentum Agent | 50 | Trend-following strategy traders |
| LLM-driven Manager Agent | 10 | AI managers guiding trade execution |
| Trade Agent | 2,950 | Execution agents under LLM manager guidance |

**(1) Liberal Day Tariffs (Institutional and Policy Factors):** On April 2, 2025, Trump announced his long-promised "reciprocal tariffs" policy, imposing a 10% baseline tax on imports from all countries, with higher rates for nations maintaining trade surpluses with the United States. This policy shock significantly affected global supply chains and multinational corporations.Before the market opened on April 4th, China proposed corresponding countermeasures to U.S. tariffs, further intensifying stock market volatility.

The simulation was conducted from April 2 to April 4. Before the market opened on April 3, we introduced pre-market news concerning the stock together with policy announcements regarding reciprocal tariffs made by former U.S. President Donald Trump. Similarly, before the opening on April 4, we incorporated pre-market news related to the stock along with official announcements from the Chinese government regarding the imposition of retaliatory tariffs. During intraday trading sessions, a selected subset of news items was released in accordance with their actual publication times.

**(2) DeepSeek Shock (Market Sentiment and Expectations):** In January 2025, Chinese company DeepSeek launched a free AI assistant claiming to use less data at a fraction of incumbent services' costs. By January 27, the assistant had overtaken ChatGPT in Apple App Store downloads, triggering a massive tech stock sell-off by global investors and causing severe market volatility.

The simulation was conducted from January 24 to January 28. Prior to market opening on January 27, we introduced pre-market news regarding the stock, coinciding with the introduction of DeepSeek into the simulation environment.

**(3) Earnings Releases (Fundamental Factors):** We select earnings announcement periods for four representative companies (AAPL, META, GOOGL, LMT) to examine market response mechanisms to fundamental information disclosure, including both quarterly and annual reports. For AAPL, META, and LMT, we use quarterly earnings releases as shock events, whereas for GOOGL, we employ its annual report.

In this simulation scenario, we focus on the release dates of annual and quarterly reports, modeling the trading days both on the announcement date and the adjacent days. Corresponding news items are introduced at the appropriate times to reflect these events.

The proportions of different participant agent types in MarketSim are determined through a hybrid process that integrates empirical evidence, established practices in market simulation, and pilot calibration. Since official data on the exact mix of market participants is not fully disclosed, we approximate the population structure using trading volume and ownership statistics. Reports indicate that retail investors contribute roughly ten percent of trading volume Adinarayan, whereas institutional investors hold approximately sixty-eight percent of equity market capitalization Brav et al. (2024). These observations motivate assigning greater capital and influence to institutional agents within the simulator. We also follow common practice in prior multi-agent simulators such as ABIDES Byrd et al. (2020), which employ a high proportion of noise agents to provide liquidity.

Building on these empirical and modeling priors, we performed a pilot calibration on a standard trading day prior to the shock. The calibration ensured that the simulated price dynamics exhibit realistic volatility patterns and that the value assessments generated by reasoning-capable agents remain interpretable rather than being dominated by excessive noise trading. The resulting configuration generalizes well across different stocks and shock scenarios without additional tuning, suggesting that it captures a stable approximation of real-world market composition.

Finally, MarketSim operates as an endogenous adaptive system in which the effective activity levels of agents evolve dynamically. Agents with reasoning capabilities may reduce participation under high uncertainty and increase participation when opportunities arise. This adaptive behavior acts as a self-regulating mechanism, making the overall dynamics less sensitive to small variations in the initial population ratios and supporting consistent behavior across a range of market environments.

## C    STATISTICAL METRICS FOR PRICE SERIES EVALUATION

To quantitatively assess the similarity between simulated and real price series, we employ seven statistical metrics that capture different aspects of time series similarity. These metrics provide a comprehensive evaluation framework for comparing the performance of price simulation models.

- **Root Mean Squared Error (RMSE)**: This measures the average magnitude of prediction errors between simulated and real prices. It is calculated as:

$$\text{RMSE} = \sqrt{\frac{1}{n}\sum_{i=1}^{n}(P_i^{\text{real}} - P_i^{\text{sim}})^2} \tag{5}$$

where $P_i^{\text{real}}$ is the real price at time $i$, $P_i^{\text{sim}}$ is the simulated price at time $i$, and $n$ is the total number of observations. Lower RMSE values indicate better simulation accuracy, with 0 representing perfect prediction.

- **Mean Absolute Percentage Error (MAPE)**: This measures the average percentage deviation between simulated and real prices, providing a scale-independent measure of accuracy:

$$\text{MAPE} = \frac{100}{n}\sum_{i=1}^{n}\left|\frac{P_i^{\text{real}} - P_i^{\text{sim}}}{P_i^{\text{real}}}\right| \tag{6}$$

MAPE values are expressed as percentages, where lower values indicate better performance. A MAPE of 10% means the simulated prices deviate from real prices by an average of 10%.

- **Dynamic Time Warping (DTW) Distance**: This captures morphological similarity by finding the optimal alignment between sequences, allowing for temporal shifts and distortions. The

DTW distance is computed using dynamic programming:

$$\text{DTW}(X, Y) = \min_{\pi} \sqrt{\sum_{(i,j) \in \pi} d(x_i, y_j)^2} \tag{7}$$

where $X = \{x_1, x_2, ..., x_m\}$ is the real price series, $Y = \{y_1, y_2, ..., y_n\}$ is the simulated price series, $\pi$ is the warping path that minimizes the cumulative distance, and $d(x_i, y_j) = |x_i - y_j|$ is the Euclidean distance between points. The warping path $\pi$ is found through the recurrence relation:

$$D(i, j) = d(x_i, y_j) + \min\{D(i-1, j), D(i, j-1), D(i-1, j-1)\} \tag{8}$$

- **Volatility Similarity Score**: This is our proposed comprehensive metric that evaluates volatility characteristic similarity across four key dimensions. The score ranges from 0 to 1, where 1 indicates perfect similarity. The four components are:

  - *Daily Volatility ($\sigma_d$)*: Calculated from percentage returns and their standard deviation, annualized for minute-level data:

  $$r_i = \frac{P_i - P_{i-1}}{P_{i-1}}, \quad \sigma_d = \text{std}(r) \times \sqrt{24 \times 60} \tag{9}$$

  where the scaling factor $\sqrt{24 \times 60}$ converts minute-level volatility to daily volatility.

  - *Volatility Frequency ($f_v$)*: Measures the proportion of returns exceeding a fixed threshold:

  $$f_v = \frac{1}{n} \sum_{i=1}^{n} \mathbb{1}(|r_i| > \tau) \tag{10}$$

  where $\tau = 0.001$ is the threshold and $\mathbb{1}(\cdot)$ is the indicator function.

  - *Peak Count ($n_{peak}$)*: Identifies local maxima using prominence-based detection with prominence threshold $\theta = 0.01 \times \sigma_P$, where $\sigma_P$ is the standard deviation of the price series.

  - *Trough Count ($n_{trough}$)*: Identifies local minima by applying peak detection to the negated price series with the same prominence threshold.

For each dimension $k \in \{\text{volatility, frequency, peaks, troughs}\}$, we calculate the relative error:

$$e_k = \begin{cases} \left| \frac{V_k^{\text{real}} - V_k^{\text{sim}}}{V_k^{\text{real}}} \right| & \text{if } V_k^{\text{real}} \neq 0 \\ 0 & \text{if } V_k^{\text{real}} = V_k^{\text{sim}} = 0 \\ 1 & \text{if } V_k^{\text{real}} = 0, V_k^{\text{sim}} \neq 0 \end{cases} \tag{11}$$

The similarity score for each dimension is $s_k = \max(0, 1 - e_k)$, and the final Volatility Similarity Score is:

$$\text{Volatility Similarity Score} = \frac{1}{4} \sum_{k=1}^{4} s_k \tag{12}$$

## D   RESULTS OF QUALITATIVE REALISM

Table S6: Stylized Facts Consistency Across All Simulation Scenarios

| STYLIZED FACTS | DeepSeek | Tariff | Earnings Releases |
|---|---|---|---|
| Absence of Linear Autocorrelation | ✓ | ✓ | ✓ |
| Fat Tails | ✓ | ✓ | ✓ |
| Aggregated Gaussianity | ✓ | ✓ | ✓ |
| Volatility Clustering | ✓ | ✓ | ✓ |
| Non-stationarity | ✓ | ✓ | ✓ |

*Note: ✓ indicates that the property is consistent with real data across all tested stocks in each scenario.*

Table S9: Performances of MarketSim and all other baselines on VST in DeepSeek Debut.

| Category | | Autoregressive | | Traditional ML | | Deep Learning | | ABM | | | | | | Ours |
|---|---|---|---|---|---|---|---|---|---|---|---|---|---|---|
| | Model | MA | ARIMA | Linear | LightGBM | LSTM | Trans. | Linear | MA | ARIMA | LightGBM | LSTM | Trans. | MarketSim |
| Qual. | Abs. of L.A. | × | × | × | ✓ | × | × | ✓ | ✓ | ✓ | ✓ | × | × | ✓ |
| | Fat Tails | ✓ | ✓ | ✓ | ✓ | ✓ | ✓ | ✓ | ✓ | ✓ | ✓ | ✓ | ✓ | ✓ |
| | Agg. Gauss. | ✓ | ✓ | × | ✓ | × | ✓ | × | ✓ | ✓ | ✓ | × | × | ✓ |
| | Volat. Clust. | ✓ | × | × | × | × | × | × | × | × | × | ✓ | ✓ | ✓ |
| | Non-Stat. | × | × | × | ✓ | × | ✓ | × | × | × | × | × | ✓ | ✓ |
| Quan. | RMSE | 42.138 | 41.404 | 45.067 | 41.676 | 62.937 | 40.657 | 45.078 | 42.148 | 41.418 | 41.683 | 62.950 | 40.645 | **23.848** |
| | MAPE (%) | 24.678 | 24.133 | 26.602 | 24.333 | 39.660 | 23.045 | 26.611 | 24.686 | 24.143 | 24.340 | 39.668 | 25.150 | **13.945** |
| | DTW Distance | 0.432 | 0.625 | 1.632 | 0.929 | 0.849 | 0.903 | 0.052 | 0.029 | 0.030 | 0.029 | 0.041 | 0.029 | **0.006** |
| | Q-Q Corr. | 0.476 | 0.543 | 0.903 | 0.706 | 0.790 | 0.880 | 0.980 | 0.975 | 0.973 | 0.982 | **0.983** | 0.982 | 0.975 |
| | Volatility Sim. | 0.015 | 0.044 | 0.015 | 0.124 | 0.060 | 0.118 | 0.041 | 0.032 | 0.036 | 0.041 | 0.155 | 0.467 | **0.539** |

## D.1 RESULTS OF QUANTITATIVE ACCURACY

Table S7: DeepSeek Simulation - Statistical Metrics

| Statistical Metrics | NVDA | VST | WMT | XOM |
|---|---|---|---|---|
| RMSE | 12.542 | 23.848 | 0.742 | 2.011 |
| MAPE (%) | 8.507 | 13.945 | 0.652 | 1.672 |
| Dynamic Time Warping Distance | 0.007 | 0.006 | 0.003 | 0.017 |
| Q-Q Correlation | 0.998 | 0.975 | 0.987 | 0.994 |
| Volatility Similarity Score | 0.540 | 0.539 | 0.593 | 0.418 |

Table S8: Earnings Releases Simulation - Statistical Metrics

| Statistical Metrics | AAPL | GOOGL | LMT | META |
|---|---|---|---|---|
| RMSE | 1.866 | 7.813 | 12.794 | 8.814 |
| MAPE (%) | 0.663 | 3.374 | 1.552 | 1.329 |
| Dynamic Time Warping Distance | 0.007 | 0.013 | 0.009 | 0.004 |
| Q-Q Correlation | 0.999 | 0.988 | 0.978 | 0.999 |
| Volatility Similarity Score | 0.674 | 0.609 | 0.435 | 0.318 |

# E RESULTS OF ABLATION STUDY

Table S10: Baseline, Risk Preferences and Different LLMs Study

| STYLIZED FACTS | Baseline | Conservative | Aggressive | Llama-3.1-8b | Qwen3-8b |
|---|---|---|---|---|---|
| Absence of Linear Autocorrelation | ✓ | ✓ | ✓ | ✓ | ✓ |
| Fat Tails | ✓ | ✓ | ✓ | ✓ | ✓ |
| Aggregated Gaussianity | ✓ | ✓ | ✓ | ✓ | ✓ |
| Volatility Clustering | ✓ | ✓ | ✓ | ✓ | ✓ |
| Non-stationarity | ✓ | ✓ | ✓ | ✓ | ✓ |
| **STATISTICAL METRICS** | | | | | |
| RMSE | 1.614 | 6.210 | 5.050 | 3.537 | 4.432 |
| MAPE (%) | 0.816 | 3.563 | 2.862 | 1.867 | 2.510 |
| Dynamic Time Warping Distance | 0.011 | 0.023 | 0.029 | 0.016 | 0.023 |
| Q-Q Correlation | 0.993 | 0.998 | 0.994 | 0.958 | 0.958 |
| Volatility Similarity Score | 0.796 | 0.597 | 0.805 | 0.441 | 0.430 |

Table S11: Strategy Ablation Study

| STYLIZED FACTS | Strategy 1 | Strategy 2 | Strategy 3 | Strategy 4 |
|---|---|---|---|---|
| Absence of Linear Autocorrelation | ✓ | ✓ | ✓ | ✓ |
| Fat Tails | × | ✓ | ✓ | ✓ |
| Aggregated Gaussianity | ✓ | ✓ | ✓ | ✓ |
| Volatility Clustering | ✓ | × | ✓ | × |
| Non-stationarity | ✓ | ✓ | ✓ | ✓ |
| **STATISTICAL METRICS** | | | | |
| RMSE | 112.579 | 1.407 | 1.620 | 2.796 |
| MAPE (%) | 59.715 | 0.751 | 0.852 | 1.432 |
| Dynamic Time Warping Distance | 0.033 | 0.013 | 0.014 | 0.026 |
| Q-Q Correlation | 0.968 | 0.950 | 0.988 | 0.998 |
| Volatility Similarity Score | 0.430 | 0.281 | 0.534 | 0.776 |

Table S12: Component Ablation Study

| STYLIZED FACTS | No Fundamental | No News | No Policy | No Reflection |
|---|---|---|---|---|
| Absence of Linear Autocorrelation | ✓ | ✓ | ✓ | × |
| Fat Tails | ✓ | ✓ | ✓ | ✓ |
| Aggregated Gaussianity | ✓ | ✓ | ✓ | ✓ |
| Volatility Clustering | × | ✓ | ✓ | ✓ |
| Non-stationarity | ✓ | ✓ | ✓ | ✓ |
| **STATISTICAL METRICS** | | | | |
| RMSE | 6.944 | 16.707 | 3.329 | 17.301 |
| MAPE (%) | 3.859 | 9.971 | 1.846 | 7.768 |
| Dynamic Time Warping Distance | 0.034 | 0.037 | 0.017 | 0.024 |
| Q-Q Correlation | 0.998 | 0.998 | 0.993 | 0.831 |
| Volatility Similarity Score | 0.781 | 0.825 | 0.775 | 0.348 |

Table S13: Liquidity Depletion Shock Study

| STYLIZED FACTS | 1% shock | 5% shock | 50% shock | 90% shock | Momentum |
|---|---|---|---|---|---|
| Absence of Linear Autocorrelation | ✓ | ✓ | ✓ | ✓ | ✓ |
| Fat Tails | ✓ | ✓ | ✓ | ✓ | ✓ |
| Aggregated Gaussianity | ✓ | ✓ | ✓ | ✓ | ✓ |
| Volatility Clustering | ✓ | ✓ | ✓ | ✓ | ✓ |
| Non-stationarity | ✓ | ✓ | ✓ | ✓ | ✓ |
| **STATISTICAL METRICS** | | | | | |
| RMSE | 3.043 | 1.888 | 1.686 | 1.817 | 2.049 |
| MAPE (%) | 1.469 | 1.054 | 0.923 | 0.991 | 1.137 |
| Dynamic Time Warping Distance | 0.026 | 0.009 | 0.009 | 0.010 | 0.012 |
| Q-Q Correlation | 0.996 | 0.999 | 0.998 | 0.997 | 0.987 |
| Volatility Similarity Score | 0.841 | 0.891 | 0.820 | 0.851 | 0.835 |

## F  MANAGER AGENT WORKFLOW

To illustrate the rationality of the manager-agent design, we will thoroughly outline their methods for market analysis. These methods involve analyzing various aspects, including news, policies, and

stock markets. The following section describes specific methodological designs that improve the interpretability of decisions made by manager agents.

### Manager Agent Workflow - Manager Agent Profile

I am a U.S. stocks short-term investment manager working for {Institution}. My investment approach follows the preferences of my company. Before trading, I will analyze market data, follow market news, and policy. My goal is to capture short-term price fluctuations within 3 to 7 trading days. The price unit in the market is cents, not dollars.

Here are some rules I must follow:

a) The amount I decide to trade should always be positive.

b) The price I need to provide is in cents.

c) My views and sentiments on market trends must be reflected through buying and selling behaviors, without considering the use of other financial instruments, such as put and call options or leverage operations.

d) Every transaction incurs transaction costs.

### Manager Agent Workflow - Market Report Agent Profile

I am a "Market Information Reporter" who needs to objectively use quantitative methods to analyze long and short forces, trend strength, support resistance levels, and capital flow after receiving the latest order book, bid and ask order depth, trading volume distribution, bid and ask spread, market depth, middle price, bid and ask strength comparison (through bid and ask depth comparison), bid and ask spread percentage and other technical indicators of a certain stock or the overall market, and express them concisely in the form of a press release.

Avoid emotional or suggestive language throughout the process to ensure information neutrality and accuracy.

Caution: I prohibit the generation of fictional content unrelated to the input data and strictly prohibit the use of input data for purposes other than relevant analysis.

### Manager Agent Workflow - News Agent Profile

I am a senior editor in the financial and political fields, working for {source}. My core task is to write accurate, in-depth, and publicly valuable reports based on the keywords provided to me, helping investors gain insight into the nature of complex events. I will focus on policy

```
changes, market trends, international relations, and other
issues, respond to sudden news quickly, and ensure that the
information is strictly verified to balance professionalism
and readability.

   Here are some rules I must follow:

   a) The news content I generate must strictly comply with
the template news, and must not indicate anywhere that the
generated news is rumors and unverified.

   b) The news content I generate must be based on the
keywords provided to me, and the generated news time must
be consistent with the input time.

   c) I will analyze the "causal chain" of events.

   d) I will not provide any personal opinions or comments.

   e) I cannot generate any data related to numbers, such as
10%.

   f) I can write with "reader thinking" and explain
professional terms in concise language.

   g) I cannot predict and report on the rise and fall of the
stock market or specific stocks.
```

**Manager Agent Workflow - Market Report**

```
   {context}

   The time of the generated report is {Generate_Time}; The
report cannot use any information after this time point.

   This is the market data list about the market:
{Market_Data_List}

   This is the technical indicator data about the present
market:  {Technical_Indicator}

   This is the trade history list:  {Trade_History}

   My task is to generate a market report using market data,
technical indicators, and trading history.

   {Temple} is the template report generated this time.

   Please generate a report in JSON format based on the
template and market data list, market technical indicators,
and trading history.  And strictly adhere to the set
character portraits without any warnings or reminders, and
are not allowed to add any explanatory text.

   Then give:  1) Market Report

   Format example:  {{'Title': '', 'Datetime':  '',
'Content':  ''}}, don't begin with any title like 'json'.

   Caution:  I cannot search for relevant data online; I can
only use the provided real data for generation.
```

**Manager Agent Workflow - News**

```
   {context}

   The time of the generated news is {Generate_Time}; The news
cannot use any information after this time point.

   This is the technical indicator data about the present
market:  {Technical_Indicator}

   This is the trade history list:  {Trade_History}

   My task is to generate internal market news based on
market technical indicators and trading history.

   {Temple} is the template news generated this time.

   Please generate a news based on the template and
technical_indicator, trade_history.  And strictly adhere
```

```
to the set character portraits without any warnings or
reminders, and are not allowed to add any explanatory text.

   Then give:  1) News

   Format example:  {{'Title':  '', 'Source':  '', 'Datetime':
'', 'content':  ''}}, don't begin with any title like 'json'.

   Caution:  I cannot search for relevant data online; I can
only use the provided real data for generation.
```

### Manager Agent Workflow - Update Fundamental Information Finance

```
   {context}

   This is my previous financial fundamental information
about {company}:  {Last_Finance_Fundamental_Information}.

   Please update my own financial fundamental information
of the {company} using one paragraph according to the
information provided(combine personal internal information
and personal investment personality), and strictly adhere
to the set character portraits without any warnings or
reminders, and are not allowed to add any explanatory text.

   Then give:  1) Fundamental information

   Format example:  {{'Finance_Fundamental_Information':  'My
financial fundamental information about XXX ...'}}, strictly
in JSON format!
```

### Manager Agent Workflow - Update Fundamental Information News

```
   {context}

   This is my previous news fundamental information of
{company}:  {Last_News_Fundamental_Information}.

   Please update my news fundamental information about
company based on the information provided, combining both
my internal insights and my investment personality.  Adhere
strictly to the specified character portraits without any
warnings or reminders, and do not include any explanatory
text.

   Then give:  1) Fundamental information

   Format example:  {{'News_Fundamental_Information':  'My
news fundamental information about XXX ...'}}, strictly in
JSON format!
```

**Manager Agent Workflow - Market Sentiment**

```
{context}

This is my investment style: {Investment_Style}.

This is my last view on market sentiment:
{Market_Sentiment}.

The current time is {Datetime}.

Here is the stock market intraday news I know:
{Intraday_News}.

Here is the present market report: {Market_News}.

This is the current market data: {Market_Data}.

Caution: I need to derive the external market sentiment
of the stock market based on non-stock market data, such
as news, and then obtain the market sentiment of the stock
market based on stock market order data.

Please analyze the market sentiment based on the
information above. Afterwards, provide the market sentiment
strictly in JSON format, combining personal internal
information and individual investment personality. Ensure
adherence to the established character profiles without any
warnings or reminders, and do not add any explanatory text.

Then give: 1) Market Sentiment

Format example: {{'External_Market_Sentiment': 'External
market sentiment for XXX ...', 'Stock_Market_Sentiment':
'Stock market sentiment for XXX ...', 'Datetime': 'Input
time'}}

Caution: In the analysis process, I provide only my
opinion on market sentiment without engaging in speculation
or predictions about stock prices.
```

**Manager Agent Workflow - Policy Indicators**

```
{context}

The company I hold shares in is {company}.

Here is the stock market intraday news/policy I know:
{Intraday_News}.

The current time is {Datetime}.
```

```
    I must objectively describe the policies and refrain from
expressing any views related to the market!

    Please analyze the institutional and policy factors
mentioned above and provide them strictly in JSON format.
Adhere to the specified character limits without including
any warnings or reminders, and do not add any explanatory
text.

    Then give:  1) Policy Indicators

    Format example:  {{'Policy_Indicators':  'Policy indicators
for XXX......', 'Datetime':  'Input time'}}
```

**Manager Agent Workflow - Technical Indicators**

```
    {context}

    The company I hold shares in is {company}.

    This is my investment style:  {Investment_Style}.

    Here is the current market data:  {Market_Data}.

    The current time is {Datetime}.

    Caution:  calculate the current stock market technical
indicator based on the current stock market order data
(The technical indicators that need to be calculated are
respectively bid ask spread, market depth, middle price, bid
ask strength comparison, bid ask spread percentage), and then
compare it with the previous market technical indicator data
to obtain your opinion on the subsequent market trend.

    Please analyse the market technical indicators above
information, then give market trend strictly in JSON
format(combine personal internal information and personal
investment personality), and strictly adhere to the set
character portraits without any warnings or reminders, and
are not allowed to add any explanatory text.

    Then give:  1) Technical Indicators

    Format example:  {{'Technical_Indicators':  'The
technical indicators in the market are respectively ...
', 'Market_Trend':  'The market trend I think is ...',
'Datetime':  'Input datetime'}}
```

**Manager Agent Workflow - Self Reflection**

```
{context}

The company I hold shares in is {company}.

This is my investment style: {Investment_Style}.

This is my surplus rate of this trade: {Surplus_Rate}

This is my surplus rate of last trade: {Last_Surplus_Rate}

This is what I think is the current stock market price:
{Last_Price}

This is my strategy for this trade: {Strategy}

The current time is {Datetime}.

Attention: My self-reflection should be divided into two
parts: strategy reflection and profit reflection, based on
profitability and chosen strategy.

Please analyze the information from the result above and
then provide a self-reflection for this trade strictly in
JSON format. Combine personal internal information and
personal investment personality. Adhere to the specified
character profiles without including any warnings or
reminders, and do not add any explanatory text.

Then give: 1) Self-Reflection

Format example: {{'Strategy_Reflection': 'My strategy
reflection for this trade ...', 'Profit_Reflection': 'My
profit reflection for this trade ...'}}
```

**Manager Agent Workflow - Update Next Goal**

```
{context}

The company I hold shares in is {company}.

This is my investment style: {Investment_Style}.

This is my self-reflection after the last round of
investment: {self_Reflection}.

Please provide the next goal for the next trade in strict
JSON format. Combine personal internal information and
personal investment personality, and strictly adhere to
the established character profiles without any warnings or
reminders. Do not include any explanatory text.
```

```
    Then give:  1) Next Goal

    Format example:  {{'Next_Goal':  'My next goal for next
trade ...'}}
```

**Manager Agent Workflow - Long Term Memory**

```
    {context}

    This is my previous long-term memory:
{Previous_Long_Term_Memory}.

    This is my short-term memory list:  {Short_Term_Memory}.

    Please give long-term memory based on my short-term
memory list and previous long-term memory, which can only
be compressed and cannot delete or ignore any information.
Return strictly in JSON format(combine personal internal
information and personal investment personality), and
strictly adhere to the set character portraits without any
warnings or reminders.

    Then give:  1) Long-Term Memory

    Format example:  {{'Long_Term_Memory':  'My long-term
memory about XXX is ...'}}.
```

**Manager Agent Workflow - Opening Price**

```
    {context}

    Based on the following information, giving the specific
opening price of the stock market in my opinion:

    a) Pay attention to the impact of news and stock market
technical indicators between the previous day's close and
today's open.

    b) Do not over-reference the trading information of the
previous day, but I can use it as a reference.

    c) I give the opening price in cents.

    This is my investment style:  {Investment_Style}

    Here are my long-term memories:

    This is the news fundamental information of {company}:
{Last_News_Fundamental_Information}.

    This is the financial fundamental information of {company}:
{Last_Finance_Fundamental_Information}.
```

```
    This is the previous institutional and policy factor:
{Previous_Institutional_Policy}.

    This is my view on the previous day market sentiment:
{Previous_Market_Sentiment}.

    This is the technical indicator data about the previous
day market: {Previous_Technical_Indicator}.

    This is my self-reflection on my previous day investment:
{Previous_Self_Reflection}.

    This is the previous trade history summary:
{Previous_Trade_History}.

    Here are my short-term memories about the current market:

    This is my view on present market sentiment:
{Market_Sentiment}.

    This is the present institutional and policy factor:
{Institutional_Policy}.

    This is my goal for this round of investment: {Next_Goal}.

    Caution: The current stock price may not truly
reflect the real value of the company at present. Please
comprehensively consider the above information, give your
opinion on the opening price, and provide the reason strictly
in JSON format. Do not return in markdown format!

    Return format example: {{'Price': '2.33', 'Reason': 'The
reason for opening price is ...'}}, don't begin with any
title like 'json'.
```

**Manager Agent Workflow - Thought Price**

```
    {context}

    Based on the following information, giving the specific
price of the stock market in my opinion, such as 10000.11,
20000.22:

    a) I need to refer to this information to provide the
specific price of the current stock market.

    b) Do not over-reference the trading information of the
previous day, but I can use it as a reference.

    c) The price I need to provide is in cents.

    This is my investment style: {Investment_Style}.
```

```
Here are my long-term memories:

This is the news fundamental information of {company}:
{Last_News_Fundamental_Information}.

This is the financial fundamental information of {company}:
{Last_Finance_Fundamental_Information}.

This is the previous institutional and policy factor:
{Previous_Institutional_Policy}.

This is my view on the previous day market sentiment:
{Previous_Market_Sentiment}.

This is the technical indicator data about the previous
day market: {Previous_Technical_Indicator}.

This is the previous trade history summary:
{Previous_trade_history}.

This is my self-reflection on my previous day investment:
{Previous_Self_Reflection}.

Here are my short-term memories about the current market:

This is the institutional and policy factor:
{Institutional_Policy}.

This is my view on market sentiment: {Market_Sentiment}.

This is the technical indicator data about the present
market: {Technical_Indicator}.

This is the last transaction price: {Last_Transaction}.

This is the trade history list: {Trade_History}.

This is my self-reflection on my last investment:
{Self_Reflection}.

This is my goal for this round of investment: {Next_Goal}.

Caution: The current stock price may not truly
reflect the real value of the company at present. Please
comprehensively consider the above information, give your
opinion on the current stock price of the company, and
provide the reason strictly in JSON format. Do not return
in markdown format!

Return format example: {{'Price': '2.33', 'Reason': 'The
reason for XXX price is ...'}}, don't begin with any title
like 'json'.
```

---

**Manager Agent Workflow - Select Strategy**

{context}

Select the most appropriate investment strategy based on the personal information, profit situation, and strategy descriptions below:

This is my investment style: {Investment_Style}.

This is my last trade's surplus rate: {Last_Surplus_Rate}

This is my view on market sentiment: {Market_Sentiment}

This is the technical indicator data about the present market: {Technical_Indicator}

My self-reflection based on last trade: {Self_Reflection}

This is my goal for this round of investment: {Next_Goal}

The following strategies are available: {Strategy_Descriptions}

Please refer strictly to my personal information and profit situation when choosing a strategy, and consider why I chose that strategy.

Return in JSON format: {{'Name': 'Strategy name', 'Reason': 'Reasons for the choice'}}

---

# G ADDITIONAL RESULTS FOR RECIPROCAL TARIFFS SCENARIO

## G.1 PRICE COMPARISON

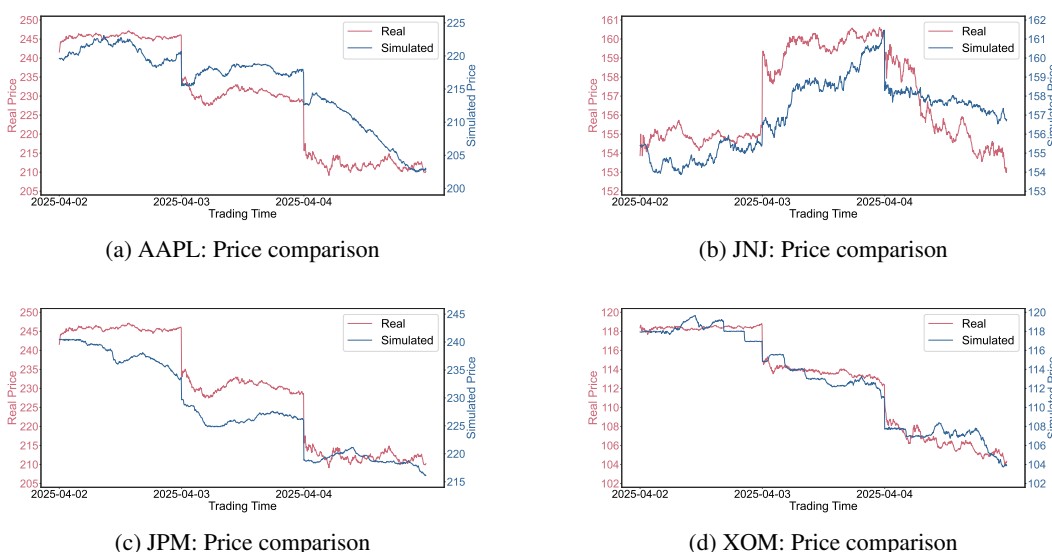

(a) AAPL: Price comparison

(b) JNJ: Price comparison

(c) JPM: Price comparison

(d) XOM: Price comparison

Figure S4: Simulated vs. real stock prices under reciprocal tariffs scenario: Price comparisons

## G.2 CANDLESTICK CHARTS

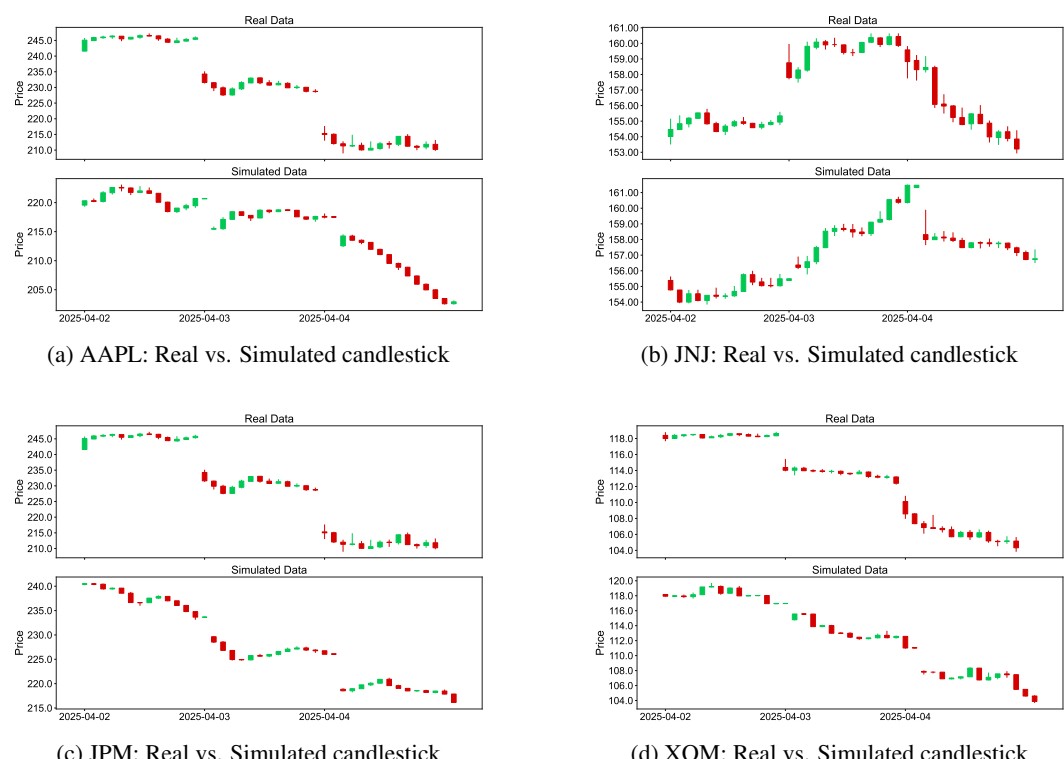

(a) AAPL: Real vs. Simulated candlestick

(b) JNJ: Real vs. Simulated candlestick

(c) JPM: Real vs. Simulated candlestick

(d) XOM: Real vs. Simulated candlestick

Figure S5: Stock price patterns under reciprocal tariffs scenario: Candlestick charts comparison

### G.3 STYLIZED FACTS

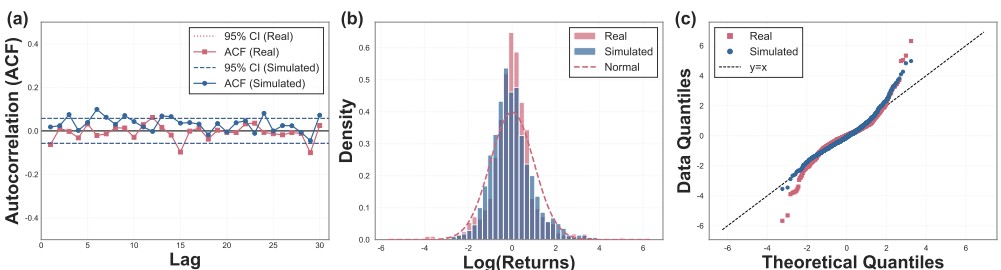

Figure S6: Simulated vs. real APPL price under reciprocal tariffs: Stylized facts comparison.

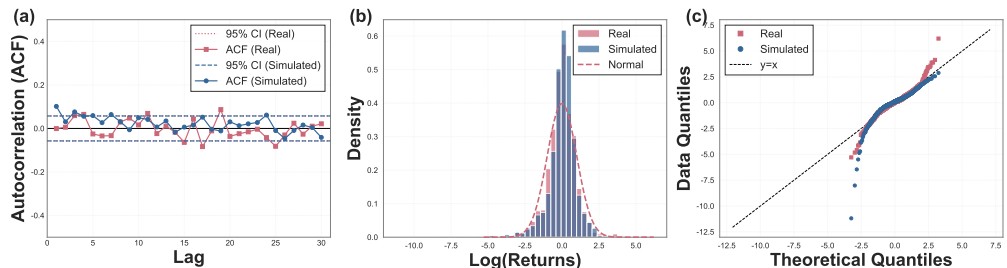

Figure S7: Simulated vs. real XOM price under reciprocal tariffs: Stylized facts comparison.

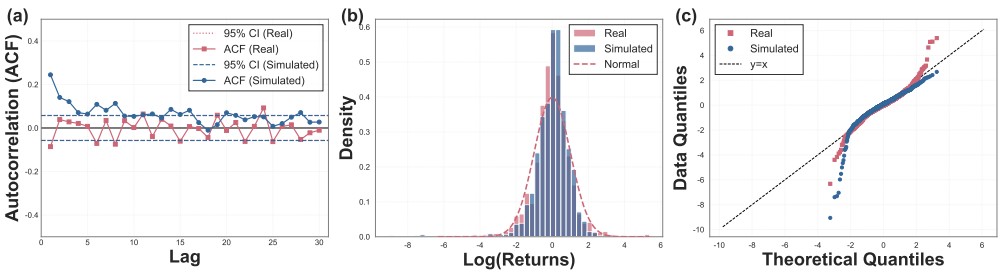

Figure S8: Simulated vs. real JPM price under reciprocal tariffs: Stylized facts comparison.

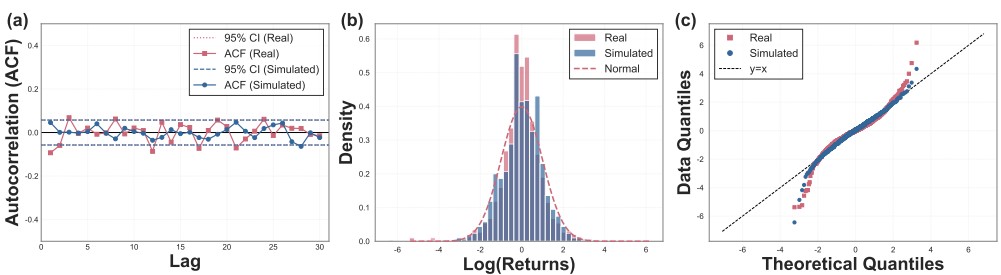

Figure S9: Simulated vs. real JNJ price under reciprocal tariffs: Stylized facts comparison.

# H    ADDITIONAL RESULTS FOR DEEPSEEK SHOCK SCENARIO

## H.1    PRICE COMPARISON

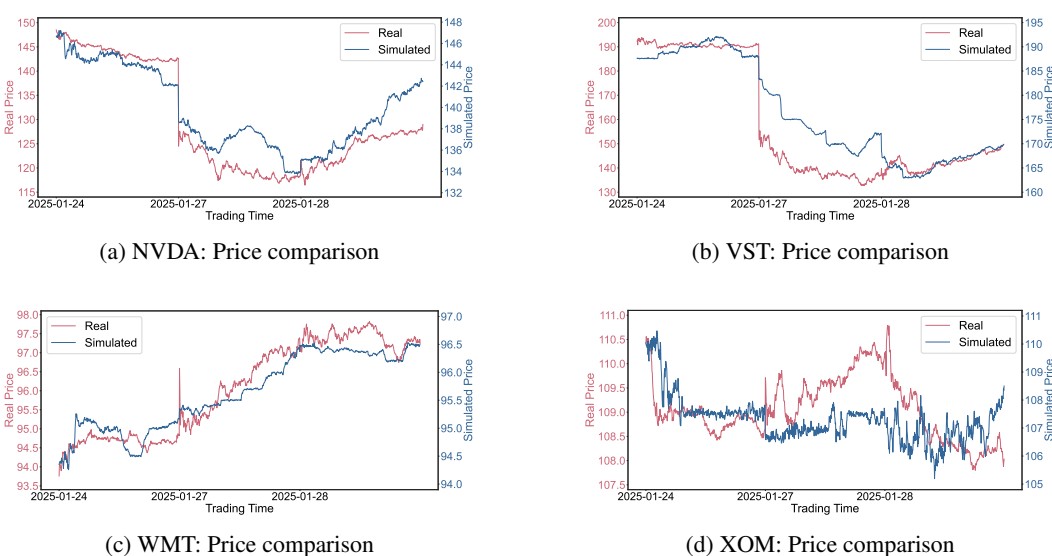

(a) NVDA: Price comparison

(b) VST: Price comparison

(c) WMT: Price comparison

(d) XOM: Price comparison

Figure S10: Simulated vs. real stock prices under DeepSeek shock: Price comparisons

## H.2    CANDLESTICK CHARTS

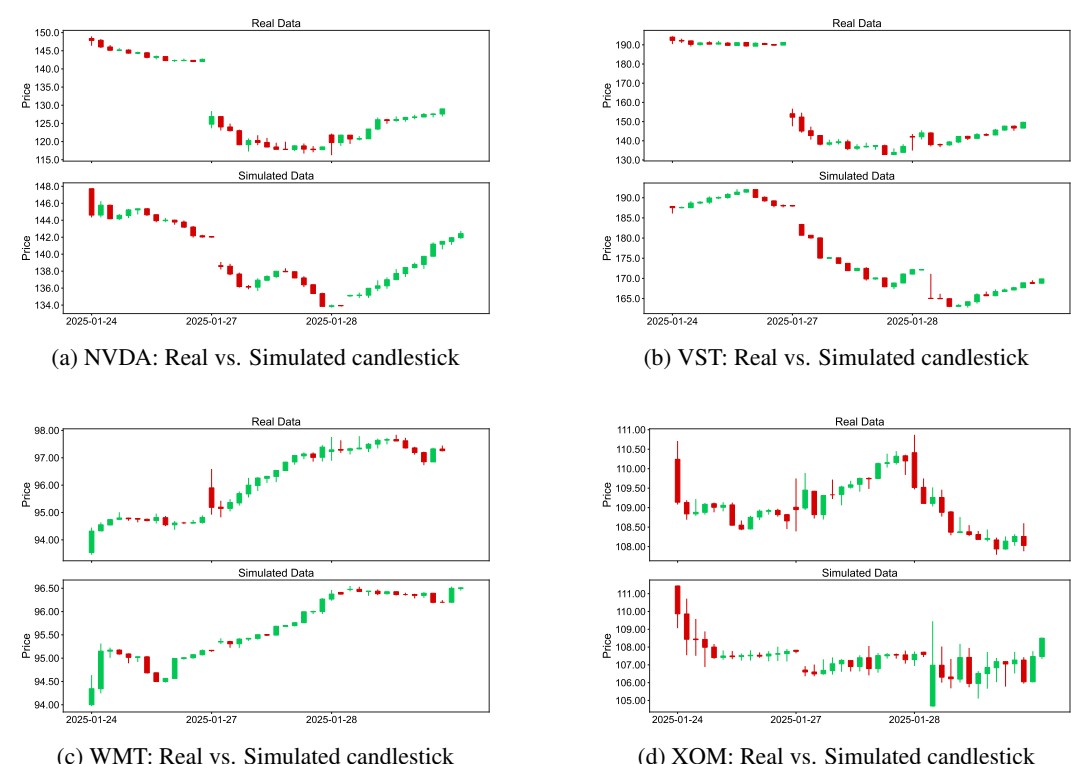

(a) NVDA: Real vs. Simulated candlestick

(b) VST: Real vs. Simulated candlestick

(c) WMT: Real vs. Simulated candlestick

(d) XOM: Real vs. Simulated candlestick

Figure S11: Stock price patterns under DeepSeek shock: Candlestick charts comparison

## H.3 STYLIZED FACTS

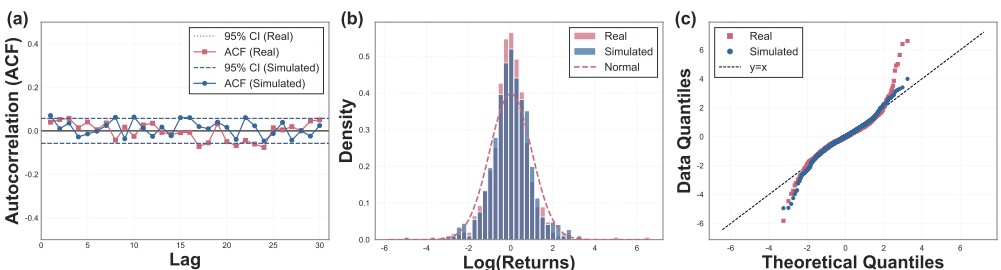

Figure S12: Simulated vs. real NVDA price under DeepSeek shock: Stylized facts comparison.

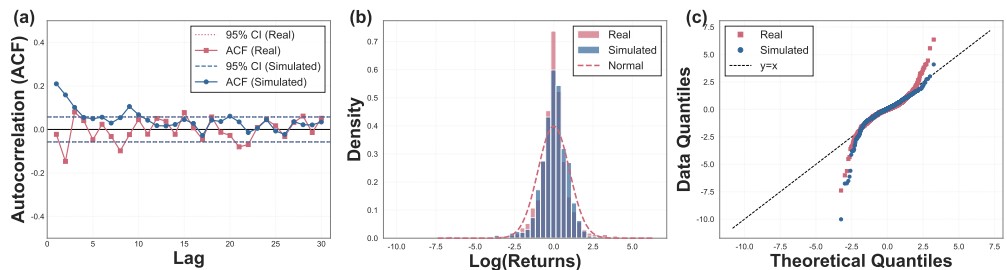

Figure S13: Simulated vs. real VST price under DeepSeek shock: Stylized facts comparison.

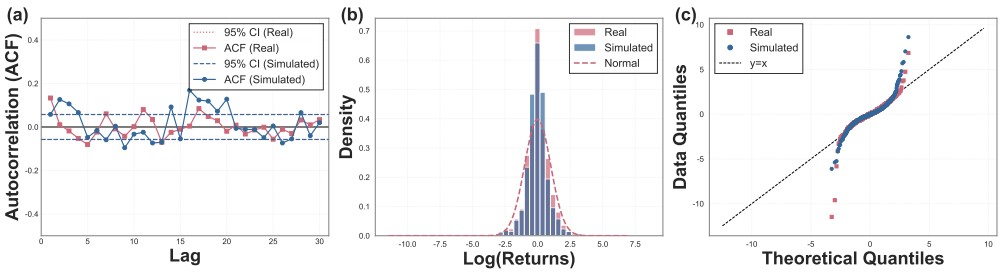

Figure S14: Simulated vs. real WMT price under DeepSeek shock: Stylized facts comparison.

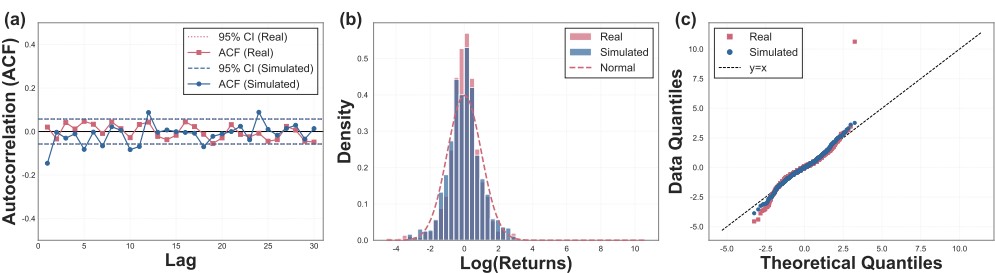

Figure S15: Simulated vs. real XOM price under DeepSeek shock: Stylized facts comparison.

# I ADDITIONAL RESULTS FOR EARNINGS RELEASES SCENARIO

## I.1 PRICE COMPARISON

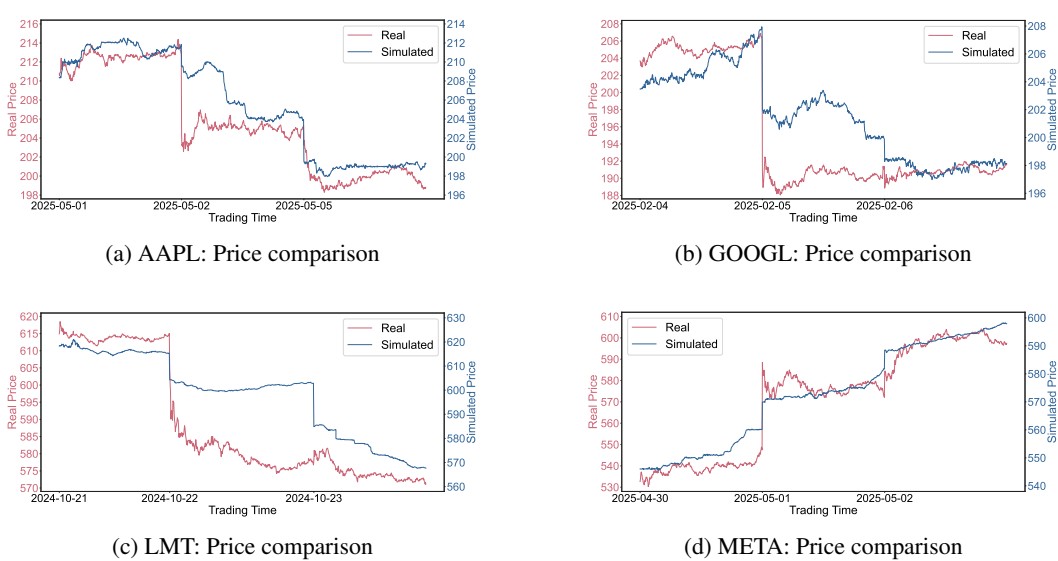

(a) AAPL: Price comparison

(b) GOOGL: Price comparison

(c) LMT: Price comparison

(d) META: Price comparison

Figure S16: Simulated vs. real stock prices under earnings releases: Price comparisons

## I.2 CANDLESTICK CHARTS

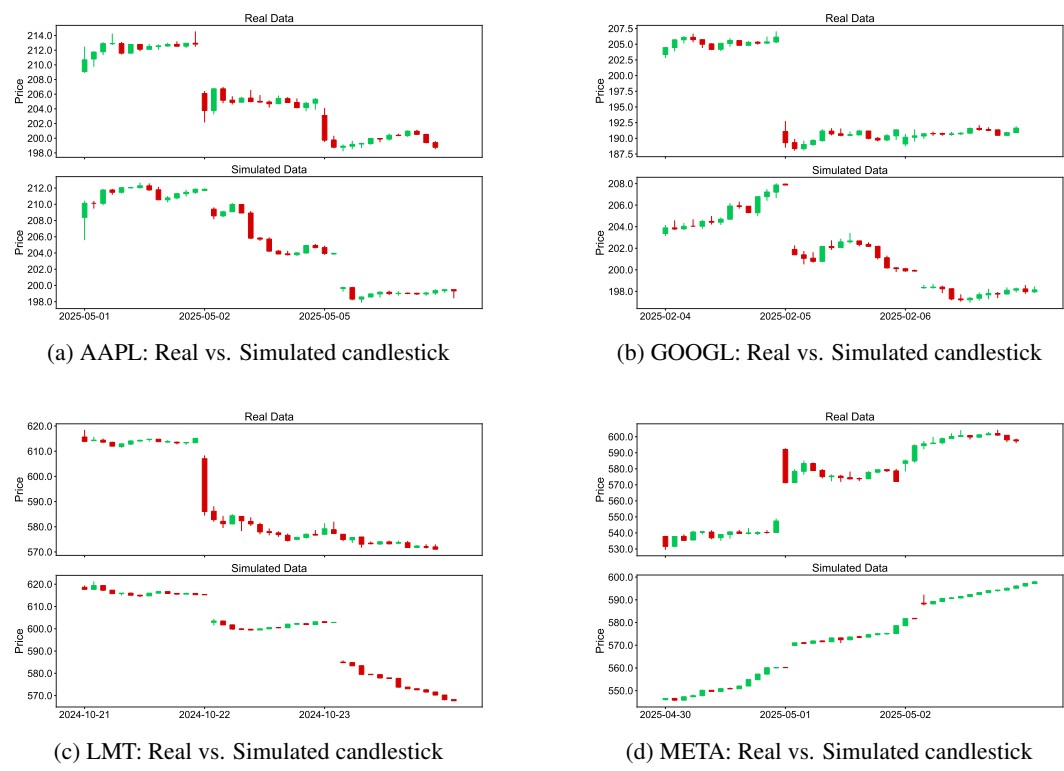

(a) AAPL: Real vs. Simulated candlestick

(b) GOOGL: Real vs. Simulated candlestick

(c) LMT: Real vs. Simulated candlestick

(d) META: Real vs. Simulated candlestick

Figure S17: Stock price patterns under earnings releases: Candlestick charts comparison

## I.3 STYLIZED FACTS

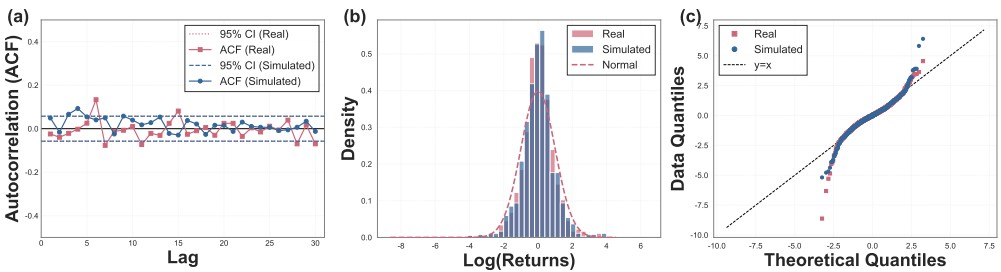

Figure S18: Simulated vs. real AAPL price under earnings releases: Stylized facts comparison.

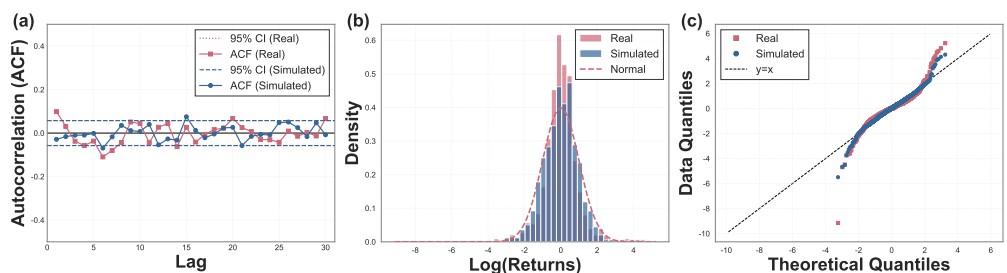

Figure S19: Simulated vs. real Google price under earnings releases: Stylized facts comparison.

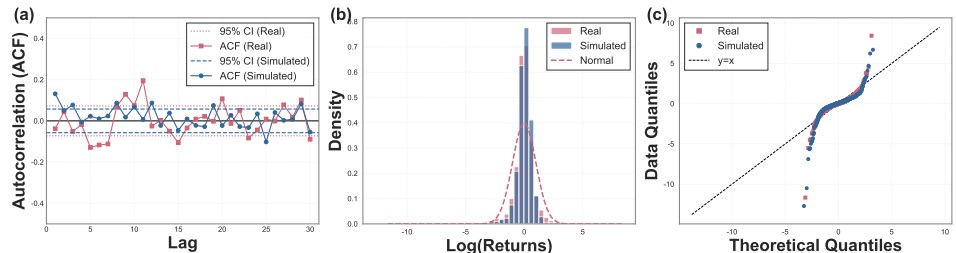

Figure S20: Simulated vs. real LMT price under earnings releases: Stylized facts comparison.

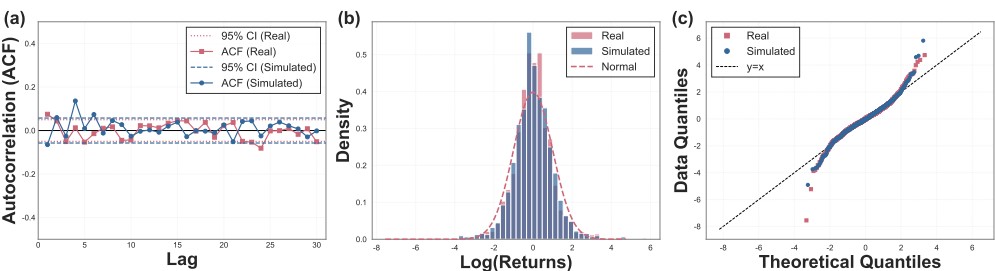

Figure S21: Simulated vs. real Meta price under earnings releases: Stylized facts comparison.

# J    BASELINE, RISK PREFERENCES AND DIFFERENT LLMS COMPARISON

## J.1    PRICE COMPARISON

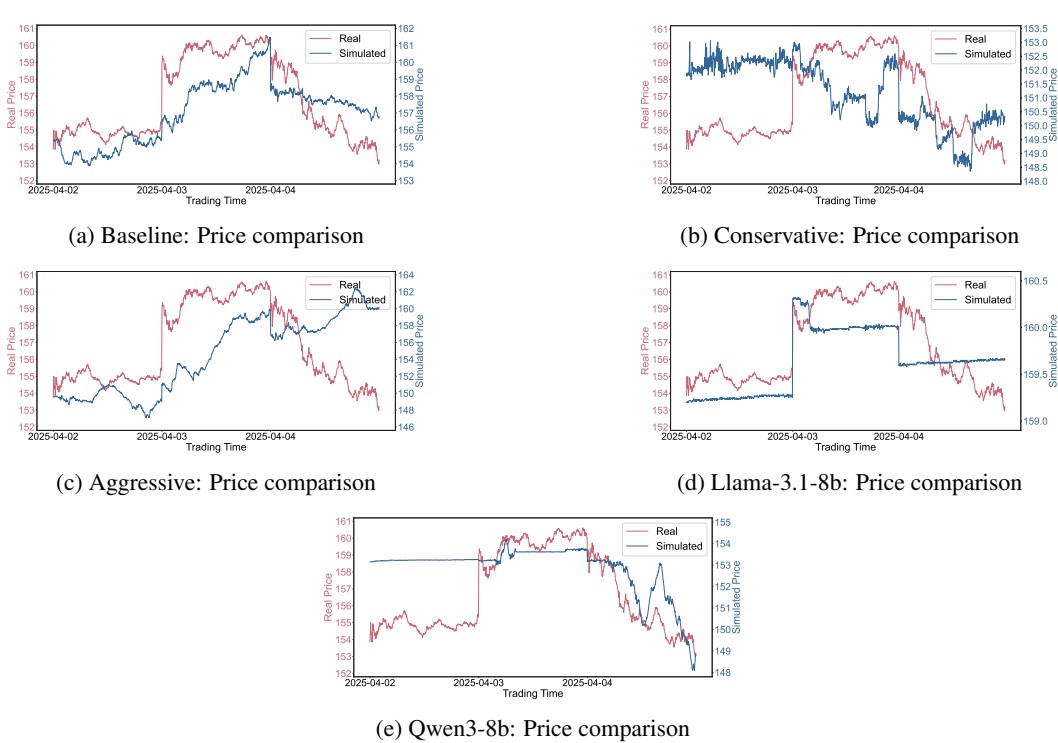

(a) Baseline: Price comparison

(b) Conservative: Price comparison

(c) Aggressive: Price comparison

(d) Llama-3.1-8b: Price comparison

(e) Qwen3-8b: Price comparison

Figure S22: Baseline, risk preferences and different LLMs: Price comparisons

## J.2    CANDLESTICK CHARTS

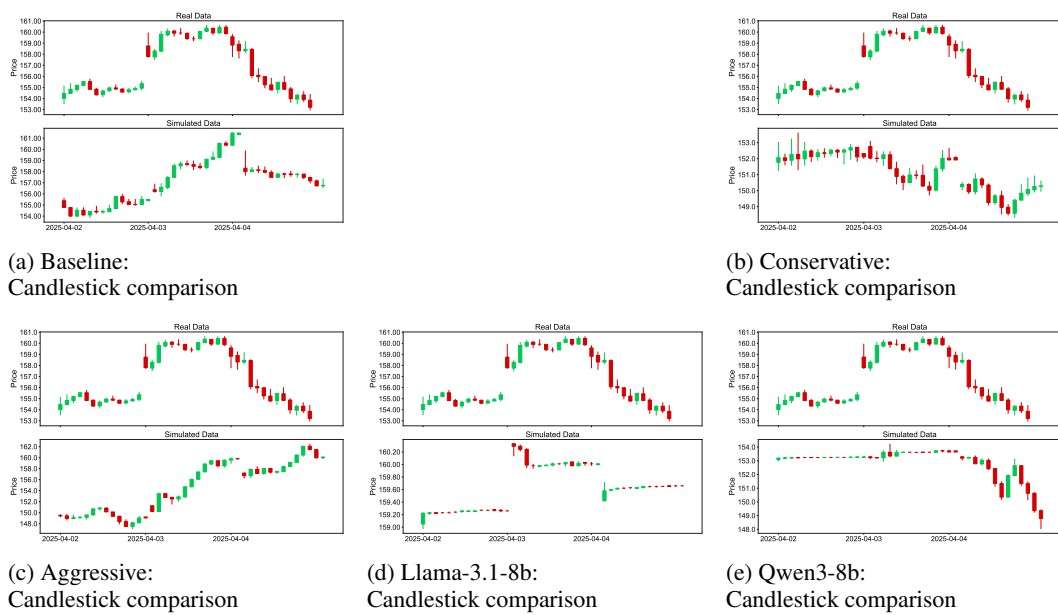

(a) Baseline: Candlestick comparison

(b) Conservative: Candlestick comparison

(c) Aggressive: Candlestick comparison

(d) Llama-3.1-8b: Candlestick comparison

(e) Qwen3-8b: Candlestick comparison

Figure S23: Baseline, risk preferences and different LLMs: Candlestick charts comparison

## J.3 STYLIZED FACTS

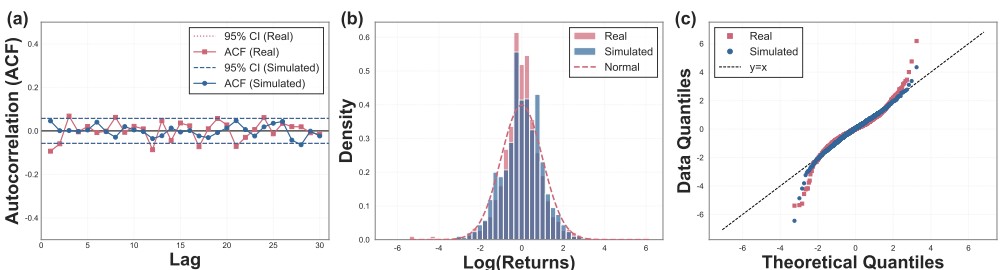

Figure S24: Baseline: Stylized facts comparison.

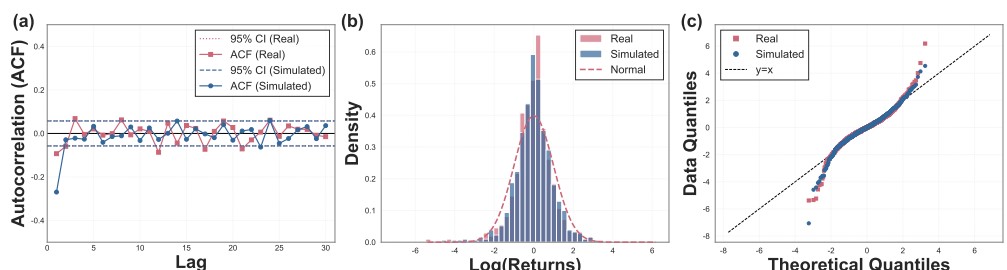

Figure S25: Conservative: Stylized facts comparison.

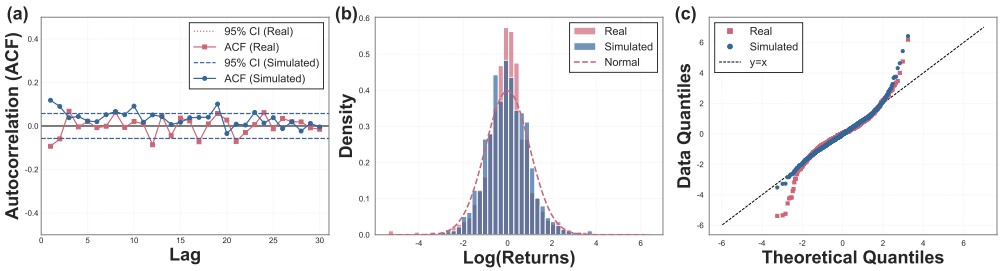

Figure S26: Aggressive: Stylized facts comparison.

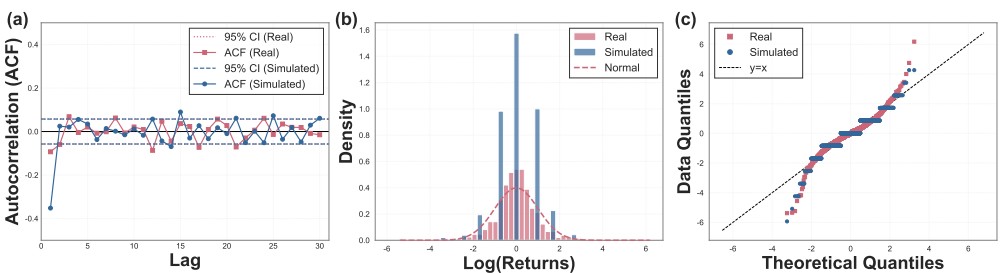

Figure S27: Llama-3.1-8b: Stylized facts comparison.

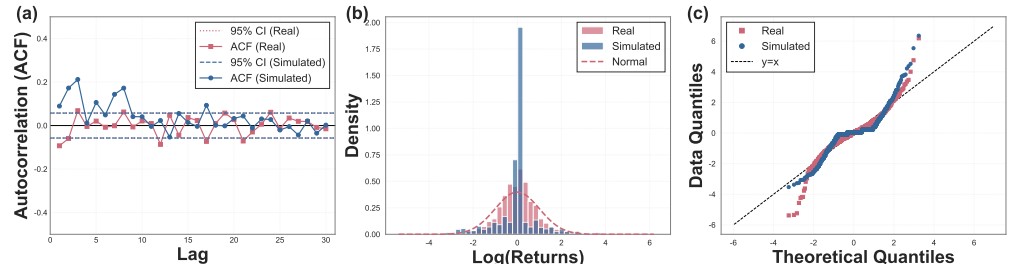

Figure S28: Qwen3-8b: Stylized facts comparison.

# K    STRATEGY COMPONENT ABLATION AND MOMENTUM STRATEGY COMPARISON

## K.1    PRICE COMPARISON

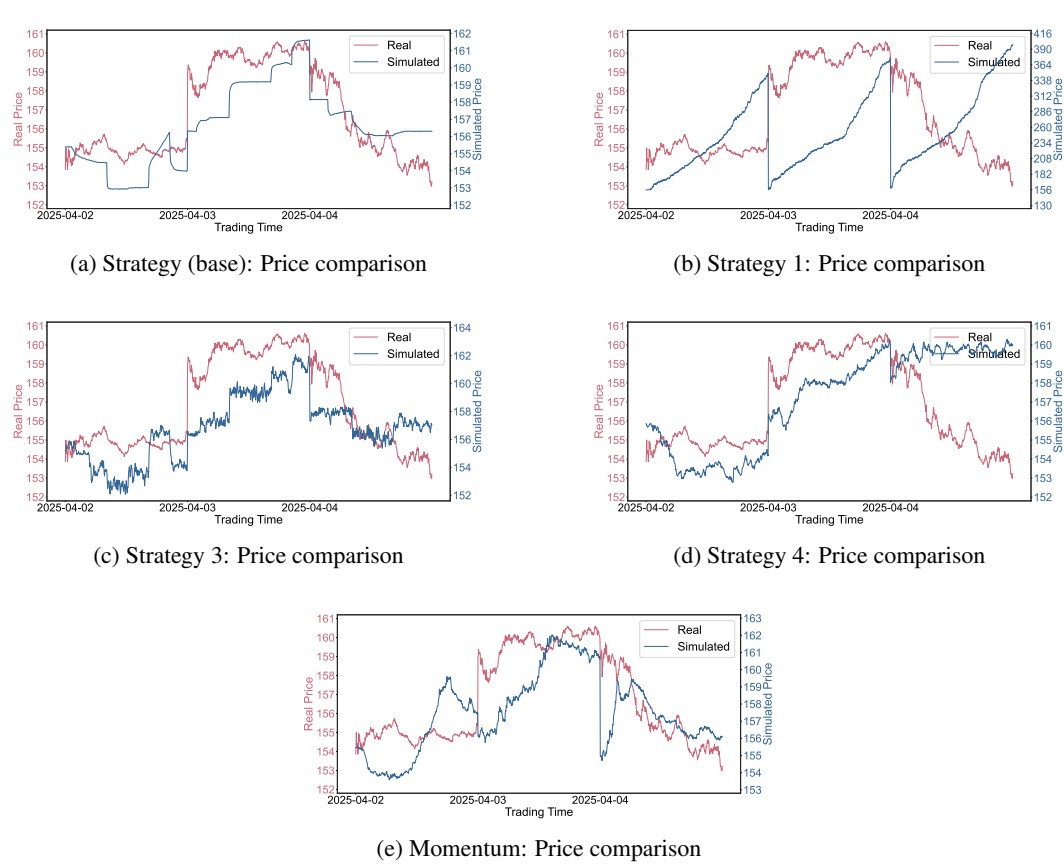

(a) Strategy (base): Price comparison

(b) Strategy 1: Price comparison

(c) Strategy 3: Price comparison

(d) Strategy 4: Price comparison

(e) Momentum: Price comparison

Figure S29: Strategy component ablation and momentum: Price comparisons

## K.2 CANDLESTICK CHARTS

(a) Strategy (base): Candlestick comparison

(b) Strategy 1: Candlestick comparison

(c) Strategy 3: Candlestick comparison

(d) Strategy 4: Candlestick comparison

(e) Momentum: Candlestick comparison

Figure S30: Strategy component ablation and momentum: Candlestick charts comparison

## K.3 STYLIZED FACTS

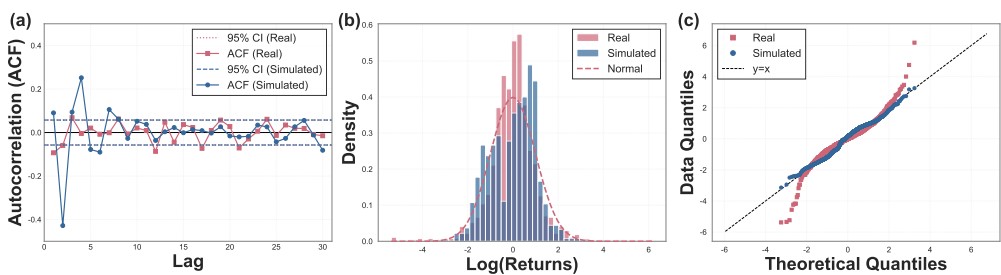

Figure S31: Strategy 1: Stylized facts comparison.

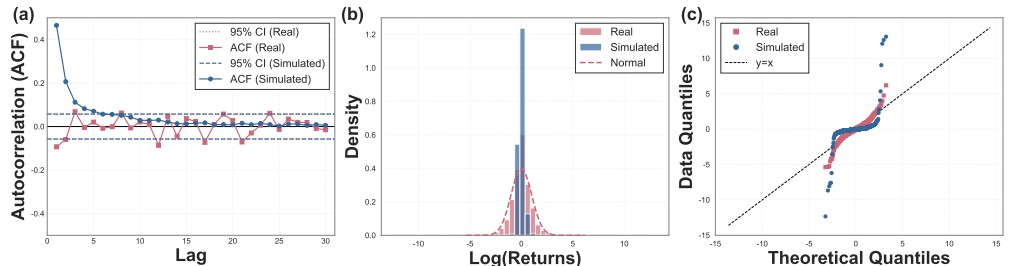

Figure S32: Strategy 2: Stylized facts comparison.

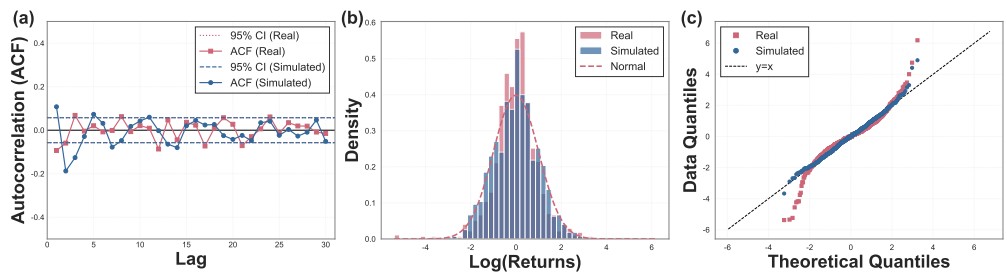

Figure S33: Strategy 3: Stylized facts comparison.

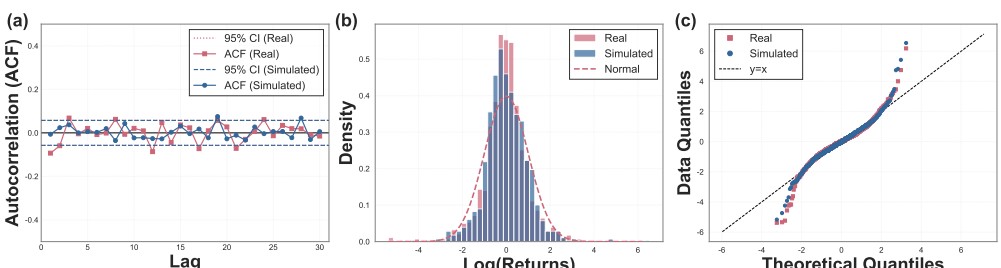

Figure S34: Strategy 4: Stylized facts comparison.

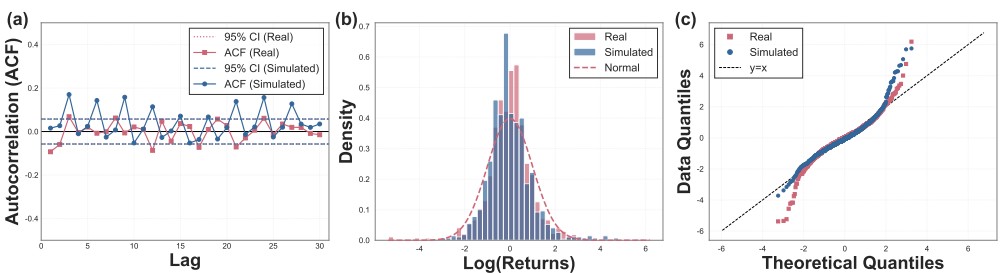

Figure S35: Momentum: Stylized facts comparison.

## L  COMPONENT ABLATION STUDY

### L.1  PRICE COMPARISON

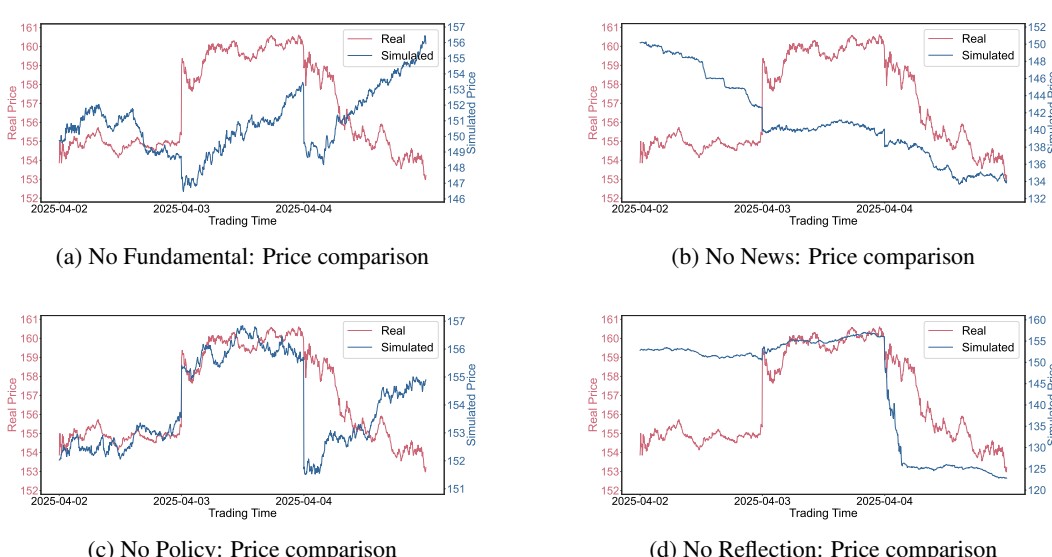

(a) No Fundamental: Price comparison

(b) No News: Price comparison

(c) No Policy: Price comparison

(d) No Reflection: Price comparison

Figure S36: Component ablation study: Price comparisons

### L.2  CANDLESTICK CHARTS

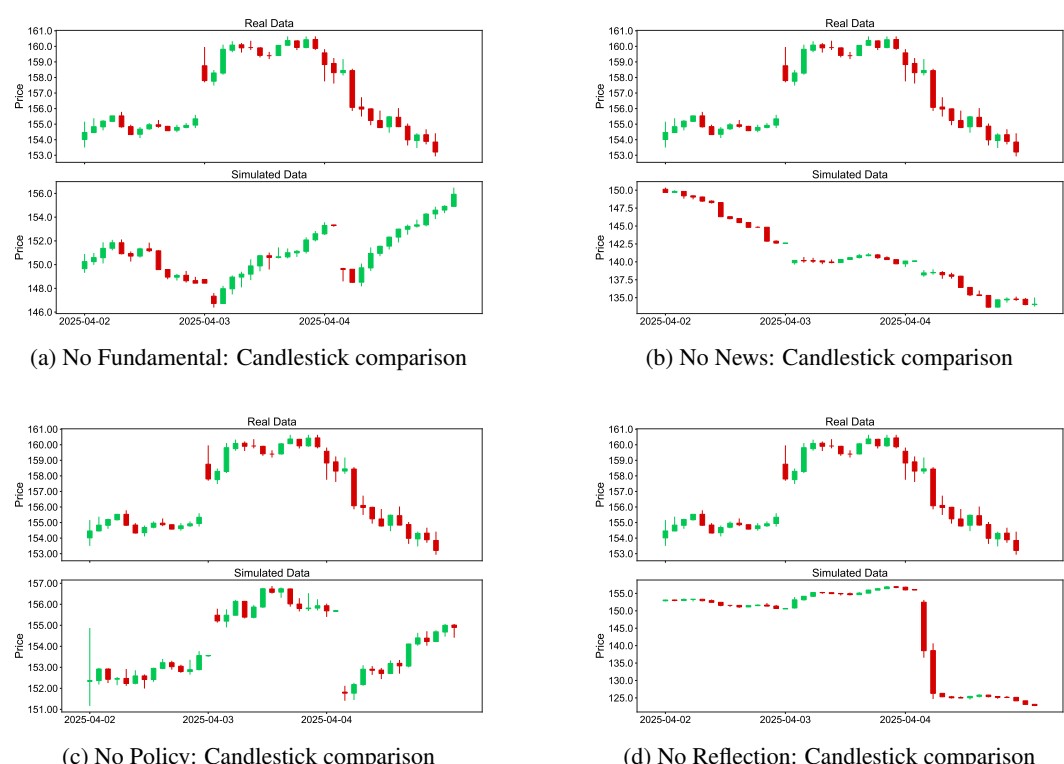

(a) No Fundamental: Candlestick comparison

(b) No News: Candlestick comparison

(c) No Policy: Candlestick comparison

(d) No Reflection: Candlestick comparison

Figure S37: Component ablation study: Candlestick charts comparison

## L.3 STYLIZED FACTS

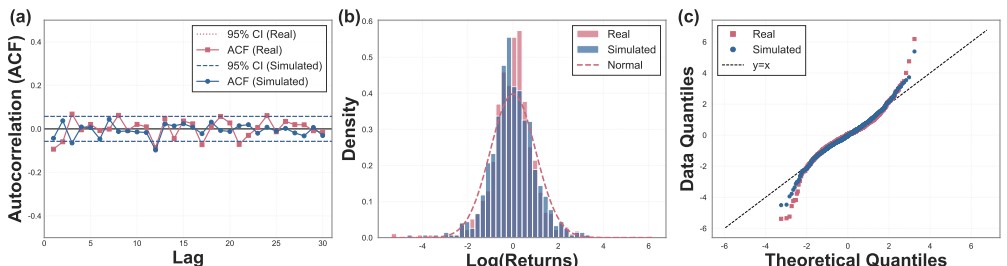

Figure S38: No Fundamental: Stylized facts comparison.

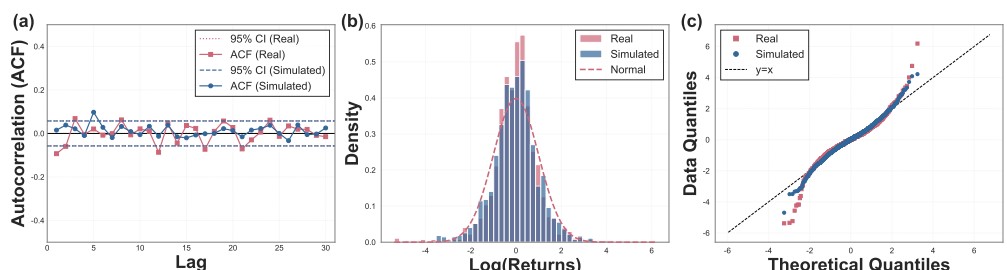

Figure S39: No News: Stylized facts comparison.

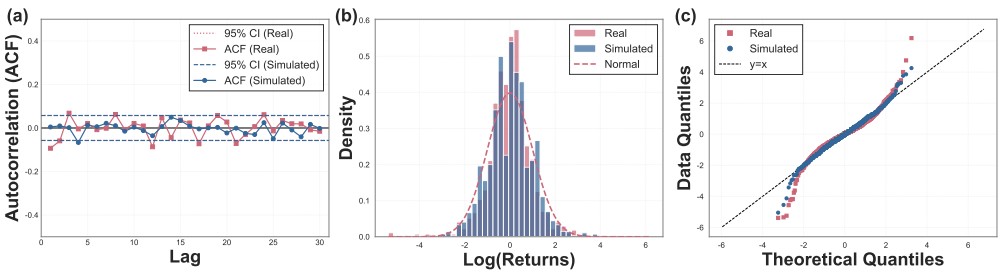

Figure S40: No Policy: Stylized facts comparison.

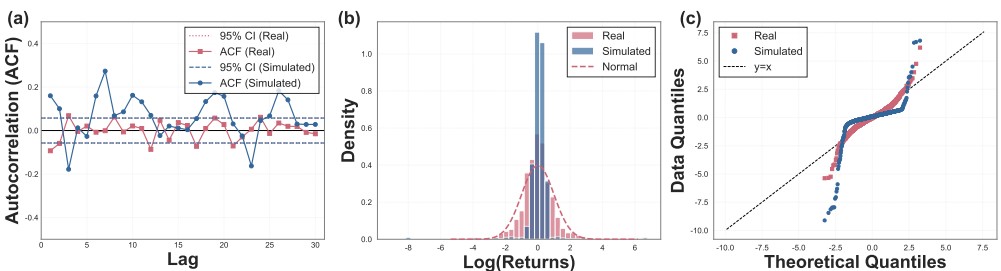

Figure S41: No Reflection: Stylized facts comparison.

# M    LIQUIDITY DEPLETION SHOCK ANALYSIS

## M.1    PRICE COMPARISON

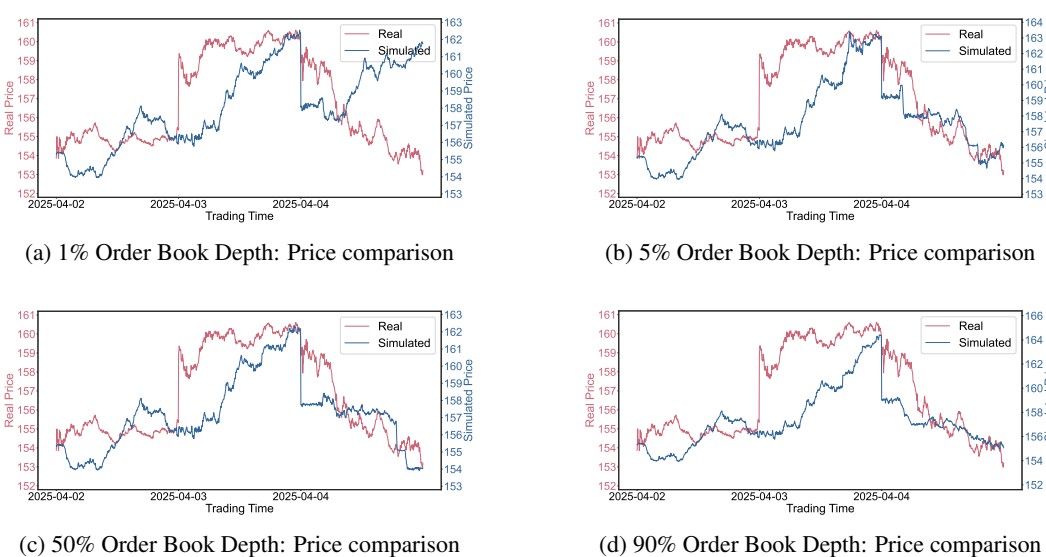

(a) 1% Order Book Depth: Price comparison

(b) 5% Order Book Depth: Price comparison

(c) 50% Order Book Depth: Price comparison

(d) 90% Order Book Depth: Price comparison

Figure S42: Liquidity depletion shock analysis: Price comparisons

## M.2    CANDLESTICK CHARTS

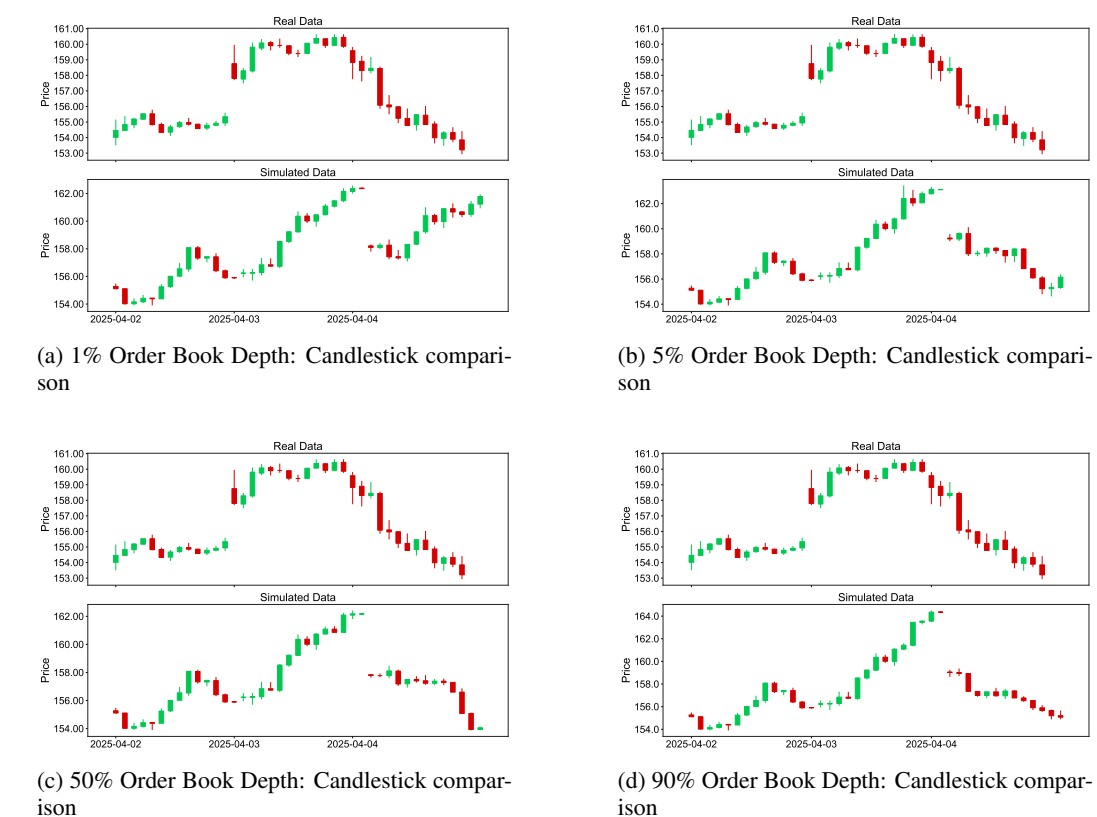

(a) 1% Order Book Depth: Candlestick comparison

(b) 5% Order Book Depth: Candlestick comparison

(c) 50% Order Book Depth: Candlestick comparison

(d) 90% Order Book Depth: Candlestick comparison

Figure S43: Liquidity depletion shock analysis: Candlestick charts comparison

M.3 STYLIZED FACTS

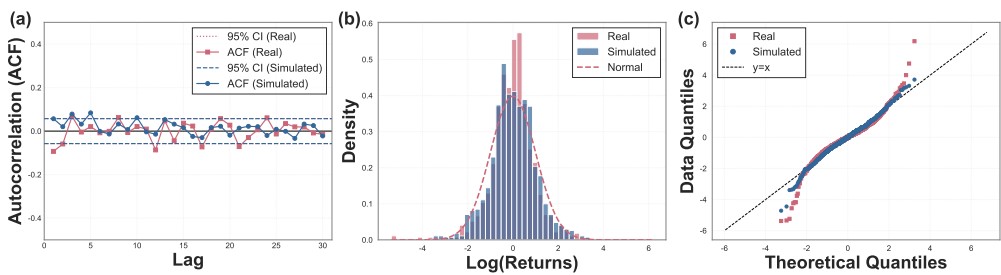

Figure S44: Simulated vs. real JNJ price under 1% Liquidity depletion shock: Stylized facts comparison.

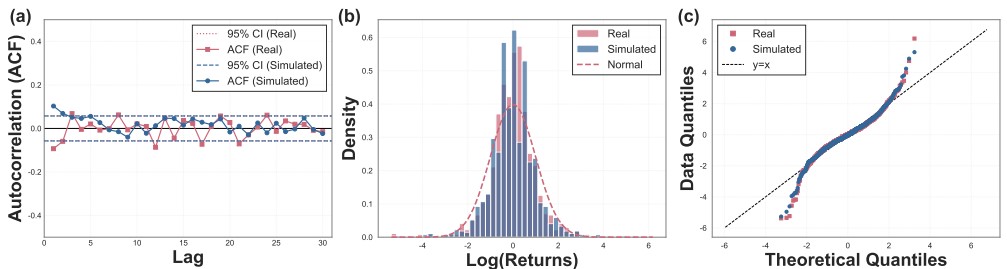

Figure S45: Simulated vs. real JNJ price under 5% Liquidity depletion shock: Stylized facts comparison.

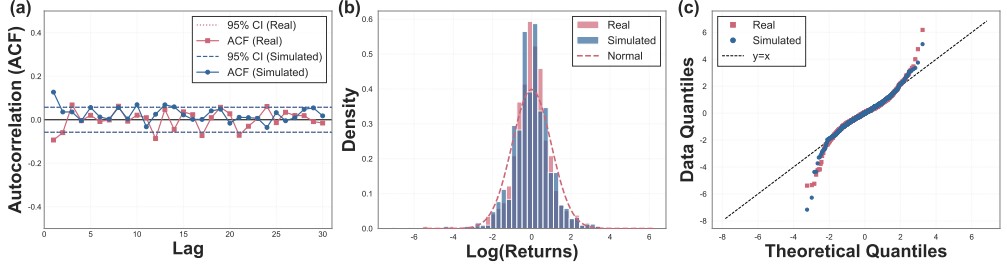

Figure S46: Simulated vs. JNJ price under 50% Liquidity depletion shock: Stylized facts comparison.

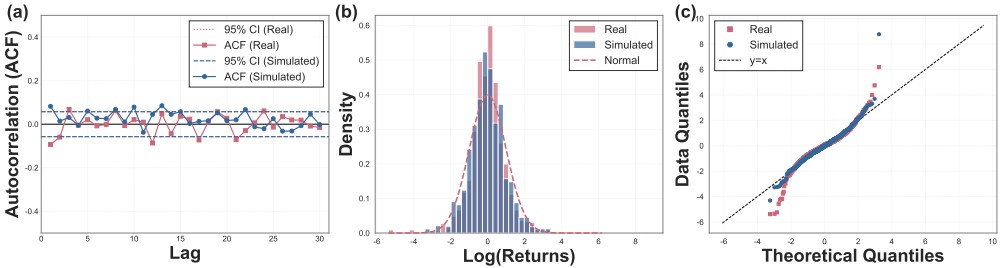

Figure S47: Simulated vs. JNJ price under 90% Liquidity depletion shock: Stylized facts comparison.

## N   PRICE DISCOVERY PROCESS

MarketSim does not use any external price inputs or anchor to any reference price. All prices emerge endogenously through a standard continuous double auction (CDA) identical to those used in real-world equity markets. Buy and sell limit orders arrive asynchronously and populate the order book

$$\mathcal{B}_t = \{(p_i^b, q_i^b)\}, \qquad \mathcal{S}_t = \{(p_j^s, q_j^s)\}.$$

The best bid and best ask are defined as

$$\text{BestBid}_t = \max_i p_i^b, \qquad \text{BestAsk}_t = \min_j p_j^s.$$

A transaction occurs when a marketable order arrives and the crossing condition

$$\text{BestBid}_t \geq \text{BestAsk}_t$$

is satisfied. The execution price follows the standard counterparty-price rule:

$$P_t = \begin{cases} \text{BestAsk}_t & \text{if a buy order crosses the book,} \\ \text{BestBid}_t & \text{if a sell order crosses the book.} \end{cases}$$

The executed volume is

$$Q_t = \min(q_{\text{best}}^b, \ q_{\text{best}}^s),$$

and the corresponding order quantities are updated.

Because $P_t$ depends solely on the internal liquidity state $(\mathcal{B}_t, \mathcal{S}_t)$, the simulator has no access to real-world prices or future information. All market dynamics arise from the endogenous interactions of heterogeneous agents operating within the CDA mechanism, ensuring that the resulting price series reflect emergent behavior rather than any form of data leakage or externally imposed reference price.

## O   COMPUTATIONAL DETAILS

All simulations run efficiently on standard CPU hardware. The experiments operate on an Intel(R) Xeon(R) Platinum 8378A CPU at 3.00GHz, without requiring any local GPU clusters. The event-driven architecture maintains a low computational load, with an average usage of 4.07 CPU cores and a peak of 11.0 cores during initialization. Once stabilized, the system operates between 2.45 and 3.20 cores, exhibiting consistent throughput.

For agent reasoning, MarketSim uses the DeepSeek API to balance performance and computational cost. A typical experiment consumes approximately 3.98 million input tokens and 0.40 million output tokens. Across the simulation horizon, the system processes 7.62 million limit orders and completes the run in approximately 4.5 hours, demonstrating that MarketSim scales to high-frequency trading workloads while remaining computationally lightweight.

