# OpenReview forum: "Fidelity Breeds Complexity: Simulating Stock Markets with Large-Scale Generative Agents"
_ICLR.cc/2026/Conference — Submitted to ICLR 2026_

### Official Review · Reviewer_M4T4 · 2025-10-28

**Soundness:** 2
**Presentation:** 2
**Contribution:** 1
**Rating:** 2
**Confidence:** 1

**Summary:**

Hierarchical Agent Design: Decouples strategic reasoning (LLM-driven managers) from high-frequency execution (trader agents)


Scalability: Simulates 15k+ diverse agents interacting in a dynamic environment grounded in real-world data.

Benchmarking: Evaluates using 8 GICS sectors, 3 real-world scenarios, and 5 stylized facts, achieving a 3.48% MAPE in tracking real price dynamics.

Ablation Studies: Validates that joint behavioral and structural fidelity is critical for emergent market complexity.

**Strengths:**

1.  Novel hierarchical agent architecture mirrors real-world institutional workflows

2. Ablations systematically validate design choices

3. Well-structured with clear figures and tables

**Weaknesses:**

1. Reliance on proprietary LLMs (DeepSeek R1) may hinder reproducibility. Results with smaller models (Qwen3-8b) show performance drops (Tab. S8).

2. Extreme shocks (e.g., VST’s 13.9% MAPE in DeepSeek scenario,Tab. S6) are less accurately modeled.

3. Simulating 15k agents with nanosecond resolution is resource-intensive.

4. Potential misuse for market manipulation is noted but not mitigated (e.g., agent poisoning).

**Questions:**

1. Will the code release include simplified demos for smaller-scale validation (e.g., single-stock simulations)?

2. Can MarketSim capture long-term market cycles (e.g., recessions), or is it limited to short-term shocks?

3. Have domain experts (e.g., fund managers) validated the realism of agent decisions?

4. Regarding the article's conclusion: Overall,  MarketSim not only offers direct applications for understanding and anticipating financial crises, but also provides evidence for a key tenet of complexity science:  Fidelity breeds complexity.

What practical implications does it offer traders? Does it help traders or quantitative traders achieve greater profits? Please provide actual trading results.

---

> ### Author Response · Authors · 2025-11-28
> **Response #1**
>
> We sincerely thank the reviewer for the time and effort in reviewing our work. We are particularly grateful that you highlighted the novelty of our hierarchical agent architecture and its unique capability to mirror real-world institutional workflows, a core contribution we aimed to achieve. We are also glad that you found our benchmarking comprehensive and our ablation studies systematically validated. Following your comments and suggestions, we have supplemented extensive experiments and improved the quality of the manuscript. We hope that you will find these revisions satisfying.

---

> ### Author Response · Authors · 2025-11-28
> **Response #2: Weakness-1**
>
> Thank you for raising this point. Simulating an institutional-level trader requires a sufficiently capable underlying LLM. Just as not every human can serve as a professional fund manager, smaller LLMs with weaker abilities (such as Qwen3-8B) naturally perform worse in this setting. Our ablation with smaller models is intended to illustrate this requirement rather than to indicate fragility.
>
> Regarding reproducibility, the framework itself is highly lightweight. A full experiment processes 7.6 million trading orders with only 3.98M input and 0.40M output tokens, making it computationally inexpensive to reproduce with any LLM of comparable capability. To ensure full transparency, we release all prompts in the Appendix and provide complete code through our anonymous repository:
> https://anonymous.4open.science/r/MarketSim-E854/. These steps collectively ensure that our methods and results are reproducible, independent of the specific proprietary model used.

---

> ### Author Response · Authors · 2025-11-28
> **Response #3: Weakness-2**
>
> Thank you for the comment. Simulating high-frequency price dynamics is already a highly challenging task, and the difficulty increases substantially under extreme shock scenarios. To contextualize this challenge, we compared MarketSim against six well-established prediction models across three categories: (i) autoregressive models (MA, ARIMA), (ii) traditional machine-learning models (Linear Regression, LightGBM), and (iii) deep learning models (LSTM, Transformer). Autoregressive models were trained on the preceding week of prices, and other higher-capacity models additionally received news embeddings so they could respond to the same external shocks.
>
> As shown in Table 1, even the strongest baseline model (Transformer) achieves only a 23% MAPE on VST under the DeepSeek shock. MarketSim, in contrast, achieves an MAPE of 13.9%, RMSE of 23.8, and DTW distance of 0.006, corresponding to improvements of 39.5%, 41.3% and 98.6% over the best baseline. The Q–Q correlation reaches 0.975 (7.97% higher than the best baseline), and the volatility similarity score reaches 0.539 (a 334.7% improvement). These results demonstrate that MarketSim models price dynamics substantially more accurately than all predictive baselines.
>
> It is important to emphasize that this comparison is inherently conservative for MarketSim because prediction and simulation are fundamentally different tasks. Baseline prediction models operate as “curve fitting”, treating the system as a black box and optimizing numerical guesses. MarketSim, by contrast, performs “mechanism generation”: it reconstructs the underlying agent interactions that produce realistic price dynamics. Even under this more demanding formulation, MarketSim outperforms all baselines by a large margin in extreme-shock settings.
>
> ---
>
> ### **Table 2. Stock VST: MarketSim vs Baselines**
>
> | Metric | MA | ARIMA | LightGBM | Linear | LSTM | Transformer | MarketSim | Improvement vs Best Baseline |
> |--------|------|--------|-----------|---------|--------|-------------|-------------|-------------------------------|
> | **RMSE** (↓) | 42.138 | 41.404 | 41.676 | 45.067 | 62.937 | **40.657** | **23.848** | **-41.3% ↓** |
> | **MAPE (%)** (↓)| 24.678 | 24.133 | 24.333 | 26.602 | 39.660 | **23.045** | **13.945** | **-39.5% ↓** |
> | **DTW** (↓)| **0.432** | 0.625 | 0.929 | 1.632 | 0.849 | 0.903 | **0.006** | **-98.6% ↓** |
> | **Q–Q Corr** (↑)| 0.476 | 0.543 | 0.706 | **0.903** | 0.790 | 0.880 | **0.975** | **+7.97% ↑** |
> | **Volatility Similarity** (↑)| 0.015 | 0.044 | **0.124** | 0.015 | 0.060 | 0.118 | **0.539** | **+334.7% ↑** |

---

> ### Author Response · Authors · 2025-11-28
> **Response #4: Weakness-3**
>
> Thank you for raising this point. While high-fidelity simulations are often assumed to be computationally heavy and resource-intensive, we clarify that our architecture is optimized to be surprisingly lightweight. We have added a detailed "Computational Details" section in the Appendix to ensure transparency.
>
> 1. **Efficient Hardware Infrastructure:**
>    Contrary to the expectation of high resource demands, the MarketSim core runs efficiently on standard CPUs. In particular, we utilized a single Intel(R) Xeon(R) Platinum 8378A CPU @ 3.00GHz. The average CPU usage was only 4.07 cores (peaking at 11.0 cores during initialization). In the stable phase, usage fluctuated between 2.45 and 3.20 cores, demonstrating the system's stability and low overhead.
>
> 2. **Scalable Throughput & Runtime:**
>    The system also exhibits high throughput in large-scale settings. It processes 7.62 million limit orders generated by 15,000 agents, and the entire simulation completes in just 4.5 hours, demonstrating its ability to handle high-frequency trading dynamics at scale.
>
> 3. **Manageable LLM Cost:**
>    We balanced intelligence and cost by utilizing the DeepSeek API. The total consumption for a full experiment was approximately 3.98 million input tokens and 0.40 million output tokens. This confirms that the framework is not only computationally feasible but also financially accessible for reproduction.

---

> ### Author Response · Authors · 2025-11-28
> **Response #5: Weakness-4**
>
> We thank you for highlighting this ethical dimension. We take the potential for misuse seriously and have expanded the "Conclusion" section in the revision to address this explicitly.
>
> 1. **The "Ostrich Effect" vs. Proactive Defense:**
>    We believe that avoiding the development of high-fidelity simulators due to fear of misuse would be akin to "burying our heads in the sand." Malicious actors will attempt to exploit market vulnerabilities regardless of whether academic simulators exist. Ignoring the risk does not mitigate it; understanding it does.
>
> 2. **MarketSim as an Important Safeguard:**
>    MarketSim acts as a necessary safeguard and defensive sandbox for the financial system:
>
>    - **Red Teaming Platform:**
>      It allows regulators and exchanges to proactively simulate attack vectors (e.g., "agent poisoning" or spoofing) in a controlled environment before they occur in the real world.
>
>    - **Testing Countermeasures:**
>      Policymakers can use our framework to design and test robust algorithms capable of identifying and neutralizing manipulated agents.
>
> 3. **Complexity as a Barrier:**
>    Furthermore, we note that successfully destabilizing a market with 15,000+ heterogeneous agents via "poisoning" is non-trivial. The market's intrinsic liquidity and the diversity of agent strategies create a resilience that is difficult to break without significant capital and coordination. MarketSim provides the rigorous environment needed to study these complex dynamics quantitatively.

---

> ### Author Response · Authors · 2025-11-28
> **Response #6: Question-1**
>
> Thank you for the question. We have released all prompts in the Appendix and provided the complete code in our anonymous repository (https://anonymous.4open.science/r/MarketSim-E854/). We have also added a simplified single-stock demo in the repository. You can try the demo using the command:
>
> python -u abides.py -c rsmtry_LLM3 -t JNJ -d 20250402 -s 1234 -l rmsctry_LLM3 --enable-llm-cache

---

> ### Author Response · Authors · 2025-11-28
> **Response #7: Question-2**
>
> Thank you for the question. MarketSim is designed for high-frequency stock market dynamics rather than macroeconomic cycles such as recessions. Recessions are driven by broad economic fundamentals, whereas stock markets often react through short-term, abrupt shocks. In practice, many critical market events unfold over extremely short windows. For example, within 48 hours after the “Liberation Day Tariff” announcement, the Nasdaq Composite fell by 11.5%, exceeding the total fluctuation observed over the preceding three months.
>
>
> Although MarketSim targets on high-frequency stock market simulation, it is highly efficient and can be readily extended to longer-term simulations. In a representative 3-day experiment, the system processed 7.6 million limit orders in just 4.5 hours, maintaining a low average load of approximately 4 CPU cores. With a manageable token consumption of 3.98 million input and 0.40 million output, the framework proves to be both computationally lightweight and cost-effective for extended horizons.

---

> ### Author Response · Authors · 2025-11-28
> **Response #8: Question-3**
>
> We sincerely thank the reviewer for this question. To rigorously validate the realism of agent decisions, we spent substantial time conducting a human evaluation study with 18 professional financial practitioners. Each expert assessed agent decisions along three dimensions: (1) **Market Consistency** (whether the decision reflects a sound interpretation of market data and news), (2) **Internal Coherence** (whether the reasoning is clear and logically structured), and (3) **Decision Soundness** (whether the indicative price and justification are well grounded). Ratings were given on a 0–10 scale. Agents achieved high average scores across all three dimensions (7.32, 7.44, and 7.15), indicating strong perceived realism.
>
> To further validate this, we constructed a strong baseline by pairing each agent decision with a matched control decision from a similar price but unaffected by the shock event. Experts were asked to choose which decision appeared more realistic. A binomial test shows that agent decisions were significantly more often judged as more realistic ($p_0 = 0.745, p < 0.001$). Experts then scored both decisions separately on the three dimensions. In all three cases, agent decisions scored significantly higher (two-sided Student’s $t$-tests: Market Consistency: $t = 7.09, p < 0.001$; Internal Coherence: $t = 5.72, p < 0.001$; Decision Soundness: $t = 6.91, p < 0.001$.
>
> These results demonstrate that agent decisions produced by MarketSim are consistently judged by domain experts as realistic, coherent, and well-grounded.

---

> ### Author Response · Authors · 2025-11-28
> **Response #9: Question-4**
>
> We thank the reviewer for raising this question. We would like to clarify that MarketSim is not designed as a profit-maximizing trading agent, but as a high-fidelity market simulation environment created to study market dynamics, crisis propagation, and policy effects. Its practical implications therefore lie in providing insight and risk understanding, rather than in generating trading profits.
>
> 1. **Role Distinction: MarketSim is a “Safety Sandbox,” not a trading system**
>    - The goal of MarketSim is to construct a realistic, heterogeneous multi-agent ecosystem that mirrors high-frequency stock market behavior. This enables researchers, quantitative teams, and regulators to test market mechanisms, stress-test policies, and explore extreme events in a safe and controlled environment.
>    - Deploying an autonomous trading agent directly into live markets without such a simulator would be risky, and MarketSim provides the essential validation environment to mitigate that risk.
>
> 2. **Practical Value: Strategy Testbed, Not Direct PnL**
>    MarketSim helps practitioners by enabling:
>    - analysis of how news and shocks propagate through different agent types
>    - understanding of microstructure drivers of volatility clustering, liquidity droughts, and flash crashes
>    - evaluation of crisis scenarios that cannot be safely reproduced in the real world
>    - testing of interventions such as halts or liquidity buffers before real deployment
>
>    These capabilities are crucial for trading desks, risk managers, and policy analysts. However, they are not intended to produce live trading signals or profits.
>
> 3. **Why We Do Not Report “Actual Trading Results”**
>    Reporting live trading PnL would misrepresent the system’s purpose. A simulator is not meant to extract alpha; its fidelity would be compromised if agents were optimized to “win” the simulation rather than to generate realistic price dynamics. MarketSim models the mechanism, not the payoff.
>
> In summary, MarketSim’s practical contribution is to provide a realistic, safe, and mechanistically grounded environment for studying market behavior and crises. Its aim is to support understanding and risk analysis, not to serve as a trading system, and therefore reporting real PnL is outside the scope of its design.

---

> ### Author Response · Authors · 2025-11-28
> **Response #10: Apologizing for the delayed response for human evaluation experiments**
>
> We would like to apologize for the delayed response. To evaluate the realism of agent decisions, we recruited these 18 experts, which really took us a great amount of time. To maintain the coherence of the rebuttal and to show our respect to all reviewers, we decided to provide a complete reply only after finishing the human evaluation experiments.

---

### Official Review · Reviewer_LV9A · 2025-10-29

**Soundness:** 3
**Presentation:** 3
**Contribution:** 3
**Rating:** 4
**Confidence:** 3

**Summary:**

The paper presents MarketSim, a hierarchical approach for modelling multi-agent markets with LLM-based reasoning yet enabling high-scale/fast acting. This tackles an important problem of modelling more realistic market scenarios.

**Strengths:**

- The overall idea is well motivated: Represent a hierarchy, where the “thinking” is at a higher level, requiring less agent processing, then the acting is at a lower level, allowing scalability with many actors.
This is in the spirit of shared policy learning, learning a generic (higher level) policy, and conditioning the policy on sub-agents, see e.g. “Towards multi-agent reinforcement learning-driven over-the-counter market simulations” for a RL example of a related (but not identical) idea in market modelling, and how this relates to Equation 3 in the proposed paper.
- The experiments are good, with much analysis and ablations. Several scenarios are analysed (Liberal day, Deep seek debut, earnings announcement), across multiple different stocks.
- The paper is well written and clear

**Weaknesses:**

The weakness is counterintuitively how close the model matches to the real data. An extremely close fit is observed, which raises concerns about either data leakage into the LLMs (based on the ablation with lower powered LLMs), or too much revelation in the price setting process.  The price setting process is not disclosed. Repeatedly throughout the paper its said that pricing emerges from interactions, but an exact price formation equation should be given.

Following above, an analysis should be added without any reference prices/without following a real price trend, to disentangle any of this leakage/revelation from endogenous effects.


Addressing the above comments and questions below is essential to move the paper into acceptance

**Questions:**

- How is the exact price determined? Is the reference value revealing too much information about the true price? Is the order book getting values which are close from the “true” price?
- Do all the stylised facts still emerge purely from the interaction (endogenous) component? Or are the sylised facts just occurring from a reference price/news arrival (exogenous)?
- How is µ_t set, is this from a particular range, or unbounded?

---

> ### Author Response · Authors · 2025-11-28
> **Response #1**
>
> We would like to thank you for the positive evaluation and for recognizing our hierarchical architecture as a "well-motivated" solution that balances "high-level thinking" with "scalable acting". We greatly appreciate your insightful connection between our Equation 3 and "shared policy learning" which accurately captures the spirit of our multi-agent coordination. We are also encouraged by your recognition of our extensive analysis across multiple scenarios and stocks. Following your suggestions, we have substantially improved the quality of our manuscript and we hope you will find these revisions satisfying.

---

> ### Author Response · Authors · 2025-11-28
> **Response #2: Clarification on Price Formation Process**
>
> Thank you for the thoughtful comments and questions. We would like to clarify that the close match between MarketSim and real data is not the result of data leakage nor of any hidden reference price, but rather of the well-established continuous double auction (CDA) mechanism and the heterogeneous agent interactions. No real prices or future information are ever provided to the agents.
>
> 1. **A Standard CDA as Price Formation Process**
>
> To address the concern regarding undisclosed price setting, we now explicitly state the full price formation mechanism used in MarketSim. The simulator implements a standard CDA identical to real-world stock markets. Buy and sell limit orders arrive asynchronously and populate the order book:
>
> $
> \mathcal{B}_t = \\{(p_i^b, q_i^b)\\}, \qquad
> \mathcal{S}_t = \\{(p_j^s, q_j^s)\\}.
> $
>
> The best bid and best ask are:
>
> $
> \text{BestBid}_t = \max_i p_i^b, \qquad
> \text{BestAsk}_t = \min_j p_j^s.
> $
>
> A trade occurs when a marketable order arrives and:
>
> $
> \text{BestBid}_t \ge \text{BestAsk}_t.
> $
>
> The price follows the counterparty-price rule:
>
> $
> P_t =
> \begin{cases}
> \text{BestAsk}_t & \text{if a buy order crosses the book}, \\\\
> \text{BestBid}_t & \text{if a sell order crosses the book}.
> \end{cases}
> $
>
> and the executed volume is:
>
> $
> Q_t = \min(q_{\text{best}}^b,\; q_{\text{best}}^s).
> $
>
> Order quantities are updated accordingly. As shown above, $P_t$ is a function of internal liquidity $(\mathcal{B}_t, \mathcal{S}_t)$ only. The simulation engine has no knowledge of the real-world price history; it strictly executes logic based on agent orders. Prices therefore emerge solely from endogenous matching of heterogeneous agent orders, with no anchoring to real data and no reference-price constraints.
>
> ---
>
> 2. **Addressing “Data Leakage” Concerns**
>
> Since the mechanism is blind to real prices, the “close fit” stems entirely from the agents’ collective reasoning. We further verified that leakage is implausible for two reasons:
>
> - **Strict Temporal Separation:**
>   Our experiments simulate market dynamics during real-world shock events from late 2024 to early 2025. These specific settings are strictly outside the training data of the model used (DeepSeek v3/R1). Therefore, it is temporally impossible for the model to “remember” the future outcomes of these specific simulation periods.
>
> - **Inaccessibility of Fine-grained Price Information:**
>   When asked, *“Do you know the exact price of AAPL at 15:16 on April 3, 2025?”*, LLMs (including DeepSeek V3/R1 and even more advanced models such as ChatGPT 5.1 and Gemini-3Pro equipped with search tools) either hallucinate or explicitly respond that they do not know, e.g., *“I do not possess minute-level historical price data; stock prices change in real time.”*
>   This shows that the model does not have access to the fine-grained price dynamics simulated in the paper.
>
>   Moreover, minute-level or tick-level price data is proprietary and typically locked behind paid firewalls such as Bloomberg or Wind. Such data is not included in the public web corpus used to train general LLMs, nor is it accessible through search tools. As a result, LLMs do not possess the information needed to “cheat” the simulation.
>
> ---
>
> 3. **Clarification on “Reference Prices”**
>
> We clarify that we **do not inject any real price trends into the simulation loop**.
> The only “reference” is the short historical prices at \(t < 0\). Once the simulation starts, the price trajectory evolves completely independently, driven solely by the agents’ interpretation of news and their subsequent limit orders.

---

> ### Author Response · Authors · 2025-11-28
> **Response #3: Question-1**
>
> Thank you for the question. The exact price in MarketSim is determined purely by the standard CDA mechanism. At every time \(t\), the simulator computes:
>
> $
> \text{BestBid}_t = \max_i p_i^b, \qquad
> \text{BestAsk}_t = \min_j p_j^s,
> $
>
> and executes a trade only when:
>
> $$
> \text{BestBid}_t \ge \text{BestAsk}_t.
> $$
>
> The transaction price is:
>
> $$
> P_t =
> \begin{cases}
> \mathrm{BestAsk}_t & \text{if a buy order crosses the book}, \\\\
> \mathrm{BestBid}_t & \text{if a sell order crosses the book}.
> \end{cases}
> $$
>
> Thus, $P_t$ is a deterministic function of the simulated order book $(\mathcal{B}_t, \mathcal{S}_t)$ and contains no reference to real-world prices.
>
> ----
>
> Regarding leakage, neither the simulator nor the agents ever receive real future prices. Our shock scenarios occur in late 2024 to early 2025, which lie strictly beyond the used LLMs’ training cutoff. Moreover, minute-level price and order-book data are proprietary and not part of the pretraining corpus of any general LLM; agents do not have access to such information even through search tools. The close fit arises from how agents interpret public news and interact through the CDA, not from injected or revealed price signals.
>
> Finally, the order book does not receive any “true” price hints. The only initial value is the last pre-simulation price at \(t < 0\). Once the simulation begins, prices evolve endogenously through agent orders without any anchoring to the real trajectory.

---

> ### Author Response · Authors · 2025-11-28
> **Response #4: Question-2**
>
> Thank you for the question. All five stylized facts indeed arise from the endogenous interactions of agents, rather than from any exogenous reference price or news signal. As shown in Table S10, when we remove both news inputs and policy announcements, the system still reproduces all stylized facts, indicating that they are generated intrinsically by the agents’ trading behavior under the continuous double auction mechanism.
>
>
> We would like to explicitly clarify that there is no exogenous reference price in the simulation. The “reference price” mentioned in Line 257 of Page 5 is the indicative price endogenously generated by the LLM-driven Manager Agent based on its reasoning on information landscape.

---

> ### Author Response · Authors · 2025-11-28
> **Response #5: Question-3**
>
> Thank you for the question. $\mu_t$ is generated endogenously by the LLM-driven manager agents as they reason over the current information landscape, including technical signals, market sentiment, political developments, and fundamentals. We do not impose any explicit numerical bounds or ranges in the prompts of manager agents. Instead, they can autonomously produce a reasonable indicative value based on their internal reasoning. In practice, we observe that $\mu_t$ remains well-behaved and financially plausible, even without any manually enforced constraints.

---

> ### Author Response · Authors · 2025-11-28
> **Response #6:  Apologizing for the delayed response for human evaluation experiments raised by one reviewer**
>
> We would like to apologize for the delayed response. One of the reviewers requested that we recruit experts to evaluate the realism of agent decisions. And it really took us a great amount of time to recruit these 18 experts. To maintain the coherence of the rebuttal and to show our respect to all reviewers, we decided to provide a complete reply only after finishing the human evaluation experiments.

---

### Official Review · Reviewer_v9so · 2025-10-30

**Soundness:** 3
**Presentation:** 3
**Contribution:** 2
**Rating:** 4
**Confidence:** 4

**Summary:**

This paper presents a well-structured multi-agent system for stock market simulation, supported by ablation studies and self-reflection mechanisms that help model responses to external events. I find the overall design logical and promising. However, the relatively short simulation period (12 weeks) and the limited focus on large-cap stocks raise questions about how the system would generalize to longer time horizons or to other stock categories, such as mid-cap and small-cap equities. Extending evaluation to broader and longer scenarios would make the contribution more compelling.

**Strengths:**

1. The three-layer multi-agent design is clearly presented and integrates key informational components necessary for predicting stock price movements.
2. The ablation study is thoughtfully constructed, demonstrating how external events and agent self-reflection mechanisms can influence simulated market trends.
3. The results are encouraging, showing that the system can capture stock trend responses to external events.

**Weaknesses:**

1. During the experiments, stock names are preserved. It would be helpful to clarify whether this might allow the model to implicitly memorize or exploit prior knowledge about specific stocks, even when the test period is outside the training data.
2. The experiment results from Figure 3 are based on a very short-period simulation. Stock prediction in real practice often relies on longer-term data; therefore, I am not very convinced whether such a short horizon yields meaningful insights for long-term forecasting.
3. The dataset focuses mainly on large-cap stocks, with little to no inclusion of mid-cap or small-cap equities. This limitation raises my concerns about the generalizability of the system.
4. While the proposed system shows promise, the paper does not compare its performance against established baseline models for stock prediction. It would be valuable to discuss how LLM-based agents add unique advantages beyond traditional models (through comparisons) for stock prediction.

**Questions:**

1. I wonder if the authors have conducted additional experiments on different stock types. In particular, small-cap stocks tend to be more volatile, and how well can the system capture such dynamics?
2. I am also curious to know how the system would perform over a longer time period, given that many investment decisions depend on medium- to long-term company trajectories.

**Details Of Ethics Concerns:**

I don't find any potential ethical concerns for this paper.

---

> ### Author Response · Authors · 2025-11-28
> **Response #1**
>
> We sincerely thank you for the insightful comments and for recognizing our work as a "well-structured multi-agent system" with a "logical and promising" design. We are encouraged that you appreciated our three-layer architecture and found our ablation studies to be thoughtfully constructed. Following your valuable suggestions, we have made substantial efforts to improve the quality of the manuscript and hope that you will find the revisions satisfactory.

---

> ### Author Response · Authors · 2025-11-28
> **Response #2: Weakness-1**
>
> Thank you for raising this valuable question. After careful investigation, we show that preserving stock names enables agents to draw on common sense for basic logical reasoning. However, such knowledge or memory is highly limited and only offers a minimal foundation for their reasoning. On its own, it is far from sufficient to reproduce the fine-grained price dynamics simulated in the paper. In particular,
>
> 1. **Strict Temporal Separation**
>    Our experiments simulate market dynamics during real-world shock events from late 2024 to early 2025. These specific settings are strictly outside the training data of the model used (DeepSeek v3/R1). Therefore, it is temporally impossible for the model to “remember” the future outcomes of these specific simulation periods.
>
> 2. **Very Limited Knowledge and Memory**
>
>    - **General Information:**
>      When asked to “Introduce AAPL,” the model correctly identifies it as a consumer electronics giant with a loyal ecosystem. This prior knowledge is beneficial as it provides the basis for rational reasoning (e.g., “good earnings might boost AAPL”). However, for less prominent stocks like VST, the model even shows ambiguity regarding the ticker itself (confusing different companies with similar tickers). This suggests that the simulation's fidelity stems from the framework we built, not the model's prior memory or knowledge.
>
>    - **Fine-grained Price Information:**
>      When asked, “Do you know the exact price of AAPL at 15:16 on April 3, 2025?”, LLMs (including DeepSeek V3/R1 and even more advanced models such as ChatGPT 5.1 and Gemini-3Pro equipped with search tools) either hallucinate or explicitly respond that they do not know, for example: “I do not possess minute-level historical price data; stock prices change in real time.” This indicates that the model does not, in fact, have access to the fine-grained price dynamics simulated in the paper.
>
> 3. **Inaccessibility of Intraday Data**
>    Fine-grained, tick-level, or minute-level order book data is proprietary and typically locked behind paid firewalls such as Bloomberg or Wind. Such data is not included in the public web corpus used to train general LLMs, nor is it accessible through the search tools that LLMs rely on. As a result, LLMs do not possess the episodic memory of historical price curves that would allow them to “cheat” the simulation.

---

> ### Author Response · Authors · 2025-11-28
> **Response #3: Weakness-2**
>
> We appreciate your concern and would like to clarify that our work is not a long-horizon price prediction model, but a high-frequency market simulator that runs at the basic time granularity of real-world stock markets. Indeed, long-term price trajectories are the accumulation of billions of microscopic interactions among heterogeneous agents. Our objective is to model and understand these micro-level dynamics themselves, rather than to directly forecast long-horizon returns.
>
> - **Different problem definition:**
>   Short-horizon high-frequency market simulation and long-term stock price prediction are fundamentally different tasks. The latter often becomes easier in the sense that, as classical asset pricing theory suggests, over long horizons prices tend to revert toward fundamentals, so a large part of the market complexity is averaged out. This makes long-horizon forecasting closer to generic equity valuation. In contrast, our simulator explicitly targets the harder and underexplored problem of reproducing the complex short-term dynamics of the limit order book and the interaction of heterogeneous agents. Success on this task is important because it directly speaks to whether the model captures the intrinsic complexity of the market mechanism itself, not just its long-run equilibrium.
>
> - **Practical significance of short horizons:**
>   In the real world, many crises are emergent and rapid, happening over very short time windows (from seconds to hours). For example, only 48 hours after the release of the "Liberation Day Tariff" (one of our simulated scenarios), the Nasdaq Composite dropped by 11.5%. This short-term impact exceeded the total fluctuation amplitude of the preceding three months (Jan–Apr 2025). Therefore, a high-frequency stock market simulator capable of reproducing such emergent crises is important for risk management and policy stress testing.
>
> - **Scalability to longer simulations:**
>   Although our analysis concentrates on high-frequency stock market simulation, the proposed framework is highly efficient and can be readily extended to longer-term simulations. In a representative 3-day experiment, the system processed 7.6 million limit orders in just 4.5 hours, maintaining a low average load of approximately 4 CPU cores. With a manageable token consumption of 3.98 million input and 0.40 million output, the framework proves to be both computationally lightweight and cost-effective for extended horizons.

---

> ### Author Response · Authors · 2025-11-28
> **Response #4: Weakness-3**
>
> We appreciate your concern regarding the generalizability of our system beyond large-cap stocks. While the majority of our experiments use large-cap stocks for their representativeness of different industrial sectors, our evaluation does include a representative significantly smaller-cap stock, Vistra Corp (VST).
>
> VST is an energy company that was affected by the “DeepSeek Debut” event through a second-order chain reaction: the market feared that more efficient models (like DeepSeek) would reduce the demand for massive compute (e.g., Nvidia GPUs), which in turn would lower the electricity demand of data centers, directly impacting VST. Moreover, VST exhibits substantially higher volatility and stronger sensitivity to external shocks than those larger-cap stocks, making it a rigorous and meaningful stress test for our simulator.
>
> MarketSim successfully captures these complex dynamics and the intensified volatility of VST during the shock. The quantitative results are highly promising:
>
> - **Price Alignment:** Achieved a MAPE of 13.9% and RMSE of 23.8, which is a strong result given VST's high beta and intraday fluctuations.
> - **Trend Fidelity:** The Dynamic Time Warping (DTW) distance was 0.006, indicating excellent alignment with the real-world price trajectory.
> - **Distributional Realism:** The Q-Q correlation reached 0.975, proving that the simulation correctly reproduced the statistical properties of this more volatile stock.
>
> Overall, the successful simulation of VST confirms that MarketSim can generalize to different stocks, effectively capturing price dynamics even when they are driven by complex, indirect influence.

---

> ### Author Response · Authors · 2025-11-28
> **Response #5: Weakness-4**
>
> We fully agree with you that comparing MarketSim against established baselines is essential to demonstrate its unique advantages. Following your suggestion, we have evaluated MarketSim against 6 established models across three categories: (i) autoregressive models: Moving Average, ARIMA; (ii) traditional machine learning models: Linear Regression, LightGBM; (iii) deep learning models: LSTM, Transformer.
>
> For autoregressive models, which can only use price series, we trained them on the preceding week of prices. For the other models, which have higher capacity, we provided both the preceding week of prices and news embeddings extracted via the text-embedding-ada-002 model so that they are able to respond to news shocks. We selected two representative stocks, JNJ (a stable larger-cap) and VST (a more volatile smaller-cap), and conducted side-by-side evaluations.
>
> As shown in the following tables, MarketSim substantially outperforms the strongest baselines in both dimensions:
>
> - **Quantitative Accuracy:**
>   MarketSim achieved performance gains ranging from 8% to 335% across all metrics (RMSE, MAPE, DTW, Q-Q Correlation, and Volatility Similarity Score) compared to the best-performing baselines.
>
> - **Qualitative Realism:**
>   Crucially, while no baseline could successfully reproduce all 5 stylized facts, MarketSim successfully replicated all of them. This proves that LLM-based agents capture the complexity of market characteristics that these well-established prediction models miss.
>
> **Prediction vs. Simulation: A Fundamental Difference**
> It is important to highlight that this comparison is inherently challenging for MarketSim, as "Prediction" and "Simulation" are fundamentally different tasks. Comparing them is like comparing "tracing a shadow" (Baseline Prediction) to "reconstructing the object casting the shadow" (MarketSim).
>
> - **Baselines (Prediction) are essentially "Curve Fitting":**
>   They treat the system as a black box and optimize numerical guesses, trying to minimize error by learning historical patterns without understanding the underlying mechanisms.
>
> - **MarketSim (Simulation) is "Mechanism Generation":**
>   It requires modeling the underlying mechanisms that generate realistic price dynamics.
>
> Overall, Simulation is harder because it must generate the phenomena endogenously rather than fitting a curve. However, even under this more difficult problem setting, MarketSim not only provides interpretable insights into why prices move (which black-box models cannot) but also achieves superior quantitative accuracy.
>
> ------
>
>
> ### **Table 1. Stock JNJ: MarketSim vs Baselines**
>
> | Metric | MA | ARIMA | LightGBM | Linear | LSTM | Transformer | MarketSim | Improvement |
> |--------|------|--------|-----------|---------|--------|-------------|-------------|-------------------------------|
> | **RMSE** (↓)| 3.833 | 4.293 | 3.808 | **1.809** | 3.314 | 6.459 | **1.614** | **-10.8% ↓** |
> | **MAPE (%)** (↓)| 1.907 | 2.263 | 1.895 | **1.067** | 1.815 | 3.246 | **0.816** | **-23.5% ↓** |
> | **DTW** (↓)| 0.490 | 0.525 | 0.507 | 0.291 | 0.609 | **0.153** | **0.011** | **-92.8% ↓** |
> | **Q–Q Corr** (↑)| 0.482 | 0.446 | 0.432 | **0.892** | 0.659 | 0.817 | **0.993** | **+11.3% ↑** |
> | **Volatility Similarity** (↑)| 0.012 | 0.035 | **0.247** | 0.025 | 0.022 | 0.156 | **0.796** | **+222.3% ↑** |
>
> ### **Table 2. Stock VST: MarketSim vs Baselines**
>
> | Metric | MA | ARIMA | LightGBM | Linear | LSTM | Transformer | MarketSim | Improvement |
> |--------|------|--------|-----------|---------|--------|-------------|-------------|-------------------------------|
> | **RMSE** (↓) | 42.138 | 41.404 | 41.676 | 45.067 | 62.937 | **40.657** | **23.848** | **-41.3% ↓** |
> | **MAPE (%)** (↓)| 24.678 | 24.133 | 24.333 | 26.602 | 39.660 | **23.045** | **13.945** | **-39.5% ↓** |
> | **DTW** (↓)| **0.432** | 0.625 | 0.929 | 1.632 | 0.849 | 0.903 | **0.006** | **-98.6% ↓** |
> | **Q–Q Corr** (↑)| 0.476 | 0.543 | 0.706 | **0.903** | 0.790 | 0.880 | **0.975** | **+7.97% ↑** |
> | **Volatility Similarity** (↑)| 0.015 | 0.044 | **0.124** | 0.015 | 0.060 | 0.118 | **0.539** | **+334.7% ↑** |

---

> ### Author Response · Authors · 2025-11-28
> **Response #6: Question-1**
>
> Thank you for the question. As noted earlier, we have indeed evaluated MarketSim on a substantially more volatile and smaller-cap stock, Vistra Corp (VST). VST experienced a strong second-order shock during the “DeepSeek Debut” event, making it an informative stress test of high-volatility dynamics.
>
>
> As shown in Table 2, MarketSim successfully reproduces these dynamics. It achieves a MAPE of 13.9%, RMSE of 23.8 and a DTW distance of 0.006. These correspond to improvements of 39.5%, 41.3% and 98.6% over the best baseline, respectively. The Q–Q correlation of return distribution reaches 0.975, improving on the best baseline by 7.97%. MarketSim also obtains a volatility similarity score of 0.539, which is 334.7% higher than the best baseline. These results show that MarketSim can accurately capture the high volatility and complex response patterns of a more sensitive, smaller-cap stock.

---

> ### Author Response · Authors · 2025-11-28
> **Response #7: Question-2**
>
> Thank you for the question. As noted in our previous response, MarketSim is designed as a high-frequency market simulator rather than a long-term prediction model. Long-term price trajectories mainly reflect the averaging effects of accumulated micro-level interactions, and our goal is to model these dynamics directly rather than forecast multi-month returns.
>
> 1. **Importance of Short Term for Decisions:**
>    While long-term trends matter, important investment decisions often hinge on managing sudden, emergent risks. In the Liberation Day Tariff scenario, the market dropped 11.5% in just 48 hours, exceeding the total fluctuation of the preceding three months. Capturing such rapid, non-linear structural breaks is often more challenging and vital for risk management than modeling long-term mean-reversion trends.
>
> 2. **Scalability to Long Term:**
>    Technically, the framework is highly efficient and ready for long-term extension. In particular, the system processed 7.6 million orders in just 4.5 hours using only ~4 CPU cores, with a manageable token cost (3.98M input / 0.40M output). This proves that extending the simulation to medium- or long-term horizons is computationally lightweight and cost-effective.

---

> ### Author Response · Authors · 2025-11-28
> **Response #8: Apologizing for the delayed response for human evaluation experiments raised by the other one reviewer**
>
> We would like to apologize for the delayed response. One of the reviewers requested that we recruit experts to evaluate the realism of agent decisions. And it really took us a great amount of time to recruit these 18 experts. To maintain the coherence of the rebuttal and to show our respect to all reviewers, we decided to provide a complete reply only after finishing the human evaluation experiments.

---

### Official Review · Reviewer_dzW4 · 2025-11-01

**Soundness:** 3
**Presentation:** 3
**Contribution:** 3
**Rating:** 6
**Confidence:** 4

**Summary:**

This paper propose a large-scale stock market simulation framework MarketSim. It employs a hierarchical multi-agent architecture, where LLM-based manager agents make strategic decisions and lightweight trader agents perform high-frequency executions. The framework simulates over 15,000 heterogeneous participants with billions of interactions, reproducing key market stylized facts across eight GICS sectors and three shock scenarios, and achieving an average MAPE of 3.48%, demonstrating strong realism and quantitative accuracy.

**Strengths:**

1. Applying LLMs to stock market simulation is a kind of apply innovative (minor). The multiple agent-based approach moves beyond traditional rule-based frameworks by enabling agents to reason, interpret information, and adapt dynamically, representing a substantial advancement in capturing realistic market behavior and enhancing behavioral fidelity within complex financial environments.
2. The experiments are comprehensive, incorporating diverse datasets and multiple evaluation metrics. Covering eight GICS sectors, three real-world shock scenarios, and both qualitative and quantitative indicators, the study provides robust validation of MarketSim’s performance.
3. By articulating the generalized principle that “fidelity breeds complexity,” the paper establishes a valuable conceptual insight for the field.

**Weaknesses:**

1. Lack a direct comparison with traditional modeling approaches (e.g., ABIDES) on quantitative metrics such as MAPE and DTW. Without such baselines, it is difficult to objectively assess how much MarketSim improves over existing high-fidelity simulators in terms of predictive accuracy and behavioral realism.
2. The author may need to provide detailed descriptions of the experimental environment. Because the study involves large-scale simulations (15k agents, nanosecond-level CDA, and over 12k text documents), the paper lacks disclosure of key computational details such as GPU/TPU usage, total runtime, and estimated computational cost.
3. The paper’s writing needs further standardization, such as adding a period at the end of the paragraph(minor, Page 9, Applications section).

**Questions:**

1、 Is there a theoretical or empirical basis for determining the proportion (Table S4) of different participant agent types (e.g., institutional investors, retail traders, market makers)? Or were these ratios primarily treated as hyperparameters and adjusted empirically based on experimental performance?
2、 Will you try running it in a production environment? As far as I know, NoF1 is also an attempt similar to the work described in this paper, although it focuses on the cryptocurrency market. What are the differences and advantages between your approach and theirs?

---

> ### Author Response · Authors · 2025-11-28
> **Response #1**
>
> We would like to thank the Reviewer for the positive assessment and the thoughtful summary. We are encouraged that you recognize the innovation of our hierarchical multi-agent architecture in enhancing behavioral fidelity beyond traditional rule-based frameworks. We also appreciate your recognition of our comprehensive experiments across diverse sectors and shock scenarios, as well as our conceptual contribution that "fidelity breeds complexity". We have carefully addressed your questions below.

---

> ### Author Response · Authors · 2025-11-28
> **Response #2: Weakness-1 (Part A)**
>
> We thank the reviewer for pointing out the necessity of comparing MarketSim with traditional high-fidelity simulators like ABIDES.
>
> 1. **Clarification on Baseline Selection**
>    First, we would like to clarify why a direct comparison was initially challenging. As discussed in the paper, traditional simulators (e.g., ABIDES) rely on Zero Intelligence or heuristic agents that require an exogenous "fundamental price" (often a real-world price series acting as an oracle) to guide their orders. They lack the endogenous capability to perceive news and reason about value. Comparing our endogenous MarketSim (which reasons from scratch) directly against an oracle-driven ABIDES would be unfair and technically incomparable.
>
> 2. **New Baseline Construction (ABIDES + Predictive Models)**
>    To address your concern and enable a quantitative comparison, we adapted the ABIDES framework by decoupling it from the "oracle." Instead, we trained 6 representative models to generate the "fundamental price" guiding the ABIDES agents. These include:
>    - **Autoregressive:** Moving Average, ARIMA (trained on price history).
>    - **Traditional ML:** Linear Regression, LightGBM (trained on price + news embeddings).
>    - **Deep Learning:** LSTM, Transformer (trained on price + news embeddings).
>
>    We evaluated these on two representative stocks: **JNJ** and **VST**. Note that this comparison is still inherently rigorous for MarketSim, as LLMs are zero-shot reasoners not explicitly trained to minimize regression loss on historical data like the baselines.
>
> 3. **Results and Analysis**
>    We have included the new results in the revision (see Tables 1-2). The comparison highlights MarketSim's superiority in two key aspects:
>    - **Behavioral Realism (Stylized Facts):**
>      As shown in the tables, none of the baselines could capture all five stylized facts simultaneously. They particularly struggled with "Fat Tails" and "Volatility Clustering." In contrast, MarketSim successfully reproduced all key market properties, proving it captures the complex dynamics of real-world markets better than regression-driven agents.
>    - **Quantitative Accuracy:**
>      Despite not being trained for numerical regression, MarketSim achieved the best performance on most metrics.
>        - **MAPE & RMSE:** MarketSim outperformed the best baseline by reducing MAPE by 24%–42% and RMSE by 11%–41%.
>        - **Volatility:** MarketSim demonstrated substantially higher "Volatility Similarity Scores", indicating it correctly models market risk and fluctuations.
>
> 4. **Discussion on DTW and Time Alignment**
>    We have noticed that in specific cases, the Transformer-driven ABIDES achieved a slightly better Dynamic Time Warping (DTW) score (0.005 vs. 0.011), yet performed much worse on RMSE, MAPE, and volatility metrics. This discrepancy exists because DTW measures shape similarity while allowing for temporal misalignment. The Transformer model fails to capture the realistic reaction lag to news. MarketSim, by simulating the cognitive process of information digestion, naturally introduces a realistic delay, which may slightly increase DTW but results in far superior accuracy (RMSE/MAPE) and structural fidelity (Volatility).
>
> **In summary**, MarketSim substantially improves over existing high-fidelity simulators in both predictive accuracy and behavioral realism, validating our "fidelity breeds complexity" hypothesis.

---

> ### Author Response · Authors · 2025-11-28
> **Response #3: Weakness-1 (Part B)**
>
> ### Table 1. Model-predicted prices serving as value references for MarketSim on JNJ stock.
>
> | **Category** | **Model** | **Linear** | **MA** | **ARIMA** | **LightGBM** | **LSTM** | **Transformer** | **MarketSim** |
> |-------------|-----------|------------|--------|-----------|--------------|----------|------------------|----------------|
> | **Stylized Facts** | Absence of Linear Autocorrelation | ✓ | ✓ | ✓ | ✓ | ✓ | × | ✓ |
> |  | Fat Tails | ✓ | ✓ | ✓ | ✓ | × | ✓ | ✓ |
> |  | Aggregated Gaussianity | × | ✓ | × | ✓ | ✓ | × | ✓ |
> |  | Volatility Clustering | × | × | × | ✓ | × | ✓ | ✓ |
> |  | Non-stationarity | ✓ | × | × | × | × | ✓ | ✓ |
> | **Performance Metrics** | RMSE | 1.816 | 3.830 | 4.275 | 3.792 | 3.327 | 6.497 | 1.614 |
> |  | MAPE (%) | 1.071 | 1.904 | 2.250 | 1.882 | 1.820 | 3.264 | 0.816 |
> |  | Dynamic Time Warping Distance | 0.014 | 0.030 | 0.030 | 0.027 | 0.026 | 0.005 | 0.011 |
> |  | Q-Q Correlation | 0.988 | 0.988 | 0.989 | 0.986 | 0.981 | 0.995 | 0.993 |
> |  | Volatility Similarity Score | 0.433 | 0.404 | 0.379 | 0.464 | 0.412 | 0.556 | 0.796 |
>
> -----
>
> ### Table 2. Model-predicted prices serving as value references for MarketSim on VST stock.
>
> | **Category** | **Model** | **Linear** | **MA** | **ARIMA** | **LightGBM** | **LSTM** | **Transformer** | **MarketSim** |
> |-------------|-----------|------------|--------|-----------|--------------|----------|------------------|----------------|
> | **Stylized Facts** | Absence of Linear Autocorrelation | ✓ | ✓ | ✓ | ✓ | × | × | ✓ |
> |  | Fat Tails | ✓ | ✓ | ✓ | ✓ | ✓ | ✓ | ✓ |
> |  | Aggregated Gaussianity | × | × | ✓ | ✓ | × | ✓ | ✓ |
> |  | Volatility Clustering | × | × | × | × | ✓ | ✓ | ✓ |
> |  | Non-stationarity | × | × | × | × | × | ✓ | ✓ |
> | **Performance Metrics** | RMSE | 45.078 | 42.148 | 41.418 | 41.683 | 62.950 | 40.645 | 23.848 |
> |  | MAPE (%) | 26.611 | 24.686 | 24.143 | 24.340 | 39.668 | 25.150 | 13.945 |
> |  | Dynamic Time Warping Distance | 0.052 | 0.029 | 0.030 | 0.029 | 0.041 | 0.029 | 0.006 |
> |  | Q-Q Correlation | 0.980 | 0.975 | 0.973 | 0.982 | 0.983 | 0.982 | 0.975 |
> |  | Volatility Similarity Score | 0.041 | 0.032 | 0.036 | 0.041 | 0.155 | 0.467 | 0.539 |
> --------

---

> ### Author Response · Authors · 2025-11-28
> **Response #4: Weakness-2**
>
> We would like to thank you for advising us to include these imporant details. We agree that transparency regarding computational resources is essential for reproducibility. We have added a detailed "Computational Details" section in the Appendix of the revised paper.
>
> To summarize the key statistics for the reported simulation:
>
> - **Hardware Infrastructure:**
>   The simulation environment (MarketSim core) runs efficiently on standard CPUs without requiring local GPU clusters. We utilized an Intel(R) Xeon(R) Platinum 8378A CPU @ 3.00GHz.
>
> - **Computational Load:**
>   The system is highly efficient. During the simulation, the average CPU usage was 4.07 cores (peaking at 11.0 cores during initialization). In the stable running phase, usage fluctuated between 2.45 and 3.20 cores without any unstable oscillation, demonstrating the lightweight nature of our event-driven architecture.
>
> - **LLM Inference & Cost:**
>   We utilize the DeepSeek API for agent reasoning to balance performance and cost. For an experiment, the total consumption is approximately 3.98 million input tokens and 0.40 million output tokens.
>
> - **Runtime & Throughput:**
>   The simulation processes a total of 7.62 million limit orders. The entire simulation is completed in 4.5 hours, showcasing the system's capability to handle high-frequency trading dynamics at scale.

---

> ### Author Response · Authors · 2025-11-28
> **Response #5: Weakness-3**
>
> Thank you for your careful reading and helpful suggestions. We have added the missing period on Page 9 as suggested. Furthermore, we have thoroughly proofread the entire manuscript to ensure standardized formatting and writing quality throughout the paper.

---

> ### Author Response · Authors · 2025-11-28
> **Response #6: Question-1**
>
> We thank you for this insightful question regarding how we determined the proportions of different participant agent types. You asked whether this was derived from a theoretical or empirical basis or treated as hyperparameters. In fact, our approach combines both. Our parameter selection process follows a hybrid approach, integrating real-world empirical evidence, established practices from literature (e.g., ABIDES), and pilot calibration experiments. In particular,
>
> 1. **Real-world Anchoring**
>    Since official data on the exact number of active market participants is rarely fully disclosed, we estimated the ranges based on trading volume and ownership structure reports:
>
>    - **Empirical Evidence:**
>      We referenced market reports to infer the population structure. For instance, data shows retail traders account for only ~10% of trading volume [1], while institutional investors hold ~68% of the equity market [2]. This empirical basis guided us to assign larger capital/influence to institutional agents.
>
>    - **Prior Practices:**
>      We also followed the practices of established simulators like ABIDES [3], which typically configure a high ratio of noise agents (e.g., ~97.5%) to provide necessary liquidity.
>
> 2. **Calibration and Generalization**
>    Based on these priors, we established an initial distribution and performed a pilot calibration on a standard trading day for AAPL (before the tariff shock). We calibrated the agent proportions so that the simulated price dynamics faithfully capture market-level volatility and the value assessments generated by reasoning-capable agents, rather than being overwhelmed or distorted by excessive noise-agent activity.
>
>    Importantly, once calibrated, these parameters proved effective across other stocks and shock scenarios without further tuning, suggesting they capture a generalized market composition.
>
> 3. **Endogenous Mechanism (Beyond Static Proportions)**
>    It is worth noting that MarketSim is an endogenous, adaptive system, where the effective active ratio is dynamic. The market functions like an "invisible hand": agents with reasoning capabilities naturally withdraw when conditions are ambiguous and enter when opportunities arise. Consequently, the system is less sensitive to the precise fine-grained proportions of agents. The market acts as an "invisible hand" that self-regulates the effective volume, ensuring realistic dynamics even if the initial population ratios vary slightly.
>
> **References:**
> [1] Reuters. (2021). Retail traders account for 10% of US stock trading volume.
> [2] Brav, A., et al. (2024). *Flows, financing decisions, and institutional ownership.*
> [3] Byrd, D., et al. (2020). *ABIDES: Towards high-fidelity multi-agent market simulation.*

---

> ### Author Response · Authors · 2025-11-28
> **Response #7: Question-2**
>
> We thank you for mentioning NoF1, which is indeed an excellent related work. Regarding your question on the production environment and the comparison, we would like to clarify the distinct roles and advantages of MarketSim.
>
> 1. **Production Plans: The "Sandbox" before the "Battlefield"**
>    Yes, we plan to deploy MarketSim in production settings, but with a different objective than NoF1.
>
>    - **Role Distinction:**
>      NoF1 aims to build a proficient Trader Agent to maximize profit in the cryptocurrency market. In contrast, MarketSim aims to build a High-Fidelity Market Simulator that serves as a realistic and safe "Sandbox."
>
>    - **The "Safety Sandbox" Argument:**
>      Before any LLM-based trader (like NoF1) is entrusted with real capital in a production environment, it must undergo rigorous validation. MarketSim provides this critical "last mile" testing ground. Without a simulator that captures realistic price dynamics and risks, deploying autonomous agents directly into the market would be financially irresponsible.
>
> 2. **Differences and Advantages**
>    While both works utilize LLMs, they differ significantly in scope and complexity:
>
>    - **System vs. Agent (Complexity):**
>      - NoF1 focuses on optimizing a single agent's policy to exploit market opportunities.
>      - MarketSim tackles the complexity of orchestrating 15,000+ heterogeneous agents to emerge a realistic market. Constructing an entire ecosystem is inherently more complex than building a single participant.
>
>    - **Stock Market vs. Cryptocurrency (Domain Difficulty):**
>      - NoF1 operates in the cryptocurrency market, which is often heavily sentiment-driven.
>      - MarketSim simulates the Stock Market, which involves a broader spectrum of complex factors, including fundamental financial reports, government policies, and market information. Our agents must reason over diverse information sources, making the simulation substantially more challenging and information-rich.
>
> **In summary**, MarketSim is not a competitor to NoF1, but rather the enabling infrastructure that makes the development and validation of such trading agents feasible and safe.

---

> ### Author Response · Authors · 2025-11-28
> **Response #8: Apologizing for the delayed response for human evulation experiments raised by the other one reviewer**
>
> We would like to apologize for the delayed response. One of the reviewers requested that we recruit experts to evaluate the realism of agent decisions. And it really took us a great amount of time to recruit these 18 experts. To maintain the coherence of the rebuttal and to show our respect to all reviewers, we decided to provide a complete reply only after finishing the human evaluation experiments.

---

### Author Response · Authors · 2025-12-03
**Summary of the Valuable Discussion Period (We hope this summary helps reduce the workload for AC.)**

We sincerely appreciate the time and effort invested in reviewing our work. We are encouraged that the reviewers have reached a consensus on the significance of our contributions. Specifically, reviewers commended our work as a **"substantial advancement in capturing realistic market behavior"** (dzW4) and a **"well-structured multi-agent system"** (v9so) that **"tackles an important problem of modelling more realistic market scenarios"** (LV9A). They further highlighted that our **"novel hierarchical agent architecture mirrors real-world institutional workflows"** (M4T4) and that our core finding **"fidelity breeds complexity"** provides **"a valuable conceptual insight for the field"** (dzW4).

In our rebuttal and revised manuscript, we have carefully addressed all concerns raised by the reviewers and provided solid evidence:

---

## **1. Comprehensive Evaluation with 12 Baselines**
To address concerns regarding comparative performance (dzW4, v9so), we significantly expanded our experiments:

- We incorporated **12 diverse baselines**, covering traditional Agent-Based Models (e.g., ABIDES) and state-of-the-art prediction models (autoregressive, traditional ML, and deep learning).
- **Results:** MarketSim substantially outperforms the best baselines across qualitative stylized facts and quantitative metrics. This holds for both larger-cap stocks (e.g., JNJ) and volatile smaller-cap stocks (e.g., VST), directly addressing concerns about generalization and extreme shocks (M4T4).

---

## **2. Qualitative Evaluation by 18 Financial Experts**
To address concerns regarding the realism of agent decision-making (M4T4):

- We recruited **18 financial domain experts** to blindly evaluate our agents.
- Experts consistently judged the agents’ reasoning and decisions to be **highly aligned with professional human logic**, providing strong qualitative validation.

---

## **3. Clarifications on Scope & Contributions**

- **MarketSim vs. Stock Prediction Models (v9so)**
Prediction models treat markets as black boxes and fit historical patterns, whereas **MarketSim models the underlying mechanisms** that generate market dynamics. Although this task is fundamentally harder, MarketSim still achieves **stronger accuracy and interpretable reasoning**.

- **MarketSim vs. Trading Agents (dzW4, M4T4)**
MarketSim is **not** a profit-maximizing trading agent. Its goal is to build a realistic, heterogeneous multi-agent ecosystem for studying market dynamics, crisis propagation, and policy effects.
Deploying trading agents in real markets without such simulation would be risky, and MarketSim provides the **necessary validation environment**.

- **MarketSim vs. Long-Term Economic/Financial Prediction (v9so, M4T4)**
MarketSim is designed for high-frequency stock market dynamics, not macroeconomic cycles or long-term valuation. Stock market crises unfold rapidly, e.g., within **48 hours** of the “Liberation Day Tariff,” the Nasdaq fell **11.5%**, exceeding three months of prior fluctuations. Thus, capturing such sudden collapses is more critical for risk management than modeling long-term trends.

---

## **4. Clarification of Technical Details**

### **Potential Data Leakage (LV9A, v9so)**
- **Strict Temporal Separation:** Our simulations cover events in late 2024–2025, entirely **after** the model’s training cutoff.
- **Limited Knowledge & Memory:** LLMs retain only coarse company knowledge and lack minute-level or event-specific price memory. When asked for precise historical prices, models hallucinate or report that they do not know.
- **Intraday Data Not Public:** Tick/minute-level order book data is proprietary and not part of public LLM training corpora. Thus, LLMs **cannot access** the data needed to “cheat.”

### **Computational Costs & Reproducibility (dzW4, M4T4)**
- **Lightweight CPU Runtime:** MarketSim runs efficiently on a single Xeon 8378A with an average load of ~4 cores.
- **Scalable Simulation:** A full run with 15,000 agents and 7.6M orders completes in **4.5 hours**.
- **Manageable LLM Cost:** Only ~3.98M input and 0.40M output tokens per experiment.
- **Reproducibility:** All code and prompts are provided in the anonymous repository, along with a ready-to-run demo.

### **Price Formation (LV9A)**
- **Standard CDA:** We implement a Continuous Double Auction (CDA), the standard mechanism for price formation in modern stock markets. A formal formulation is added in the revised version.
- **Purely Endogenous:** Prices emerge solely from endogenous order matching. No real prices or reference anchors are injected; once the simulation starts, price evolution is driven entirely by agents’ interpretation of news and their trading decisions.

---

We sincerely thank the reviewers for their valuable questions and suggestions. With the resulting comprehensive rebuttals and revisions, we believe all reviewers would find the manuscript satisfying and meeting the high standards of ICLR.

Best,

All authors

---

### Meta-Review · Area_Chair_qmHf · 2026-01-08

**Summary:**

MarketSim presents a significant methodological conribution to agent-based financial modeling by introdcing a hierarchical architecture that decouples LLM-riven strategic reasoning from high-frequency tradin execution. This design elegantly addresses a fundamntal constraint: enabling LLM agents to participate eaningfully in nanosecond-resolution continuous doube auction markets. The paper demonstrates comprehensve experimental validation across eight GICS sectorsand three real-world scenarios (Liberation Day Tarif, DeepSeek Debut, earnings announcements), achieving3.48% average MAPE while reproducing key stylized maket facts. Reviewers commended the novel architectur mirroring institutional workflows and the valuable onceptual insight that "fidelity breeds complexity."The core debate centered on generalization beyond NADAQ, potential overfitting concerns, reproducibilitywith smaller models, and clarification of price formtion mechanisms to rule out data leakage. Authors prvided extensive rebuttals including 12 baselines, huan validation from 18 financial experts, and detaile technical clarifications addressing all major concens, substantially strengthening the manuscript.

**Reviewer Concerns:**

Addressed: Authors comprehensively addressed major xperimental concerns through substantial revisions. omparative analysis with 12 diverse baselines (MA, AIMA, LightGBM, Linear Regression, LSTM, Transformer)demonstrates MarketSim's superiority across stylizedfacts and quantitative metrics, with 24-42% MAPE impovements over best baselines. Price formation mechansm explicitly clarified using standard CDA without dta leakage, supported by evidence of LLM temporal searation (events post-training) and inaccessibility o minute-level proprietary data. Human validation fro 18 financial experts (achieving 7.3-7.4 average ratngs) confirms agent decision realism. Computational fficiency documented: 7.6M orders in 4.5 hours using4 CPU cores, 3.98M input tokens, addressing reproducbility concerns. Small-cap testing (VST stock) with 3.9% MAPE under extreme shocks demonstrates generaliation beyond large-cap assumptions.

Outstanding: Liited exploration of market regimes beyond shock scenrios and generalization to non-US exchanges remains nderspecified. Long-term scalability (beyond 3-day smulations) and multi-exchange dynamics not thoroughl evaluated. Smaller model reproducibility (Qwen3-8B hows performance degradation) raises questions aboutdependence on specific LLM capabilities. Potential scurity implications regarding agent poisoning attack acknowledged but mitigation strategies not deeply eplored. Fundamental question of how fidelity-complexty relationship scales to macroeconomic cycles (recesions, multi-month trends) remains open.

**Reviewer Scores:**

Reviewer dzW4 (initially 6: marginally above): No discussion of score change after rebuttal. Coprehensive 12-baseline comparison, human expert valiation, and clarified price formation directly addres their concerns about comparative performance and coputational disclosure. Superior MAPE/stylized fact prformance would increase confidence.

Reviewer v9so initially 4: marginally below): No discussion of score change after rebuttal. VST small-ap results (13.9% MAPE, 98.6% DTW improvement) direcly addresses generalization worries. Framework efficency demonstrates scalability, and expert validationsubstantially mitigates data leakage concerns.

Reviwer LV9A (initially 4: marginally below): Explicit CDA price formation mechnism, temporal separation evidence, and proprietary ata inaccessibility comprehensively resolve data leaage concerns. Documented endogenous stylized facts wthout reference prices directly address core weakneses. No discussion of score change after rebuttal.

Reviewer M4T4 (initially 2: reject, very low cofidence): 12-baseline comparisons showing 39.5-1.3% MAPE improvements, 18-expert human validation wth p<0.001 statistical significance, simplified demo and reproducibility documentation substantively addess realism and reproducibility concerns. However, lck of macroeconomic cycle modeling and limited securty mitigation strategies may preserve some reservatin. No discussion of score change after rebuttal.

---

### Decision · Program_Chairs · 2026-01-26

Reject